## Registered report

psychology/cognition

priming search, tendency to explore, domain-general exploration, explore–exploit

**Author for correspondence:**
Davide Marchiori
e-mail: davmar@sam.sdu.dk

# Priming exploration across domains: does search in a spatial environment influence search in a cognitive environment?

## Farid Anvari and Davide Marchiori

Strategic Organization Design Group, Department of Marketing and Management, University of Southern Denmark, Odense 5230, Denmark

(iD) FA, 0000-0002-5806-5654

Is there a general tendency to explore that connects search behaviour across different domains? Although the experimental evidence collected so far suggests an affirmative answer, this fundamental question about human behaviour remains open. A feasible way to test the domain-generality hypothesis is that of testing the so-called priming hypothesis: priming explorative behaviour in one domain should subsequently influence explorative behaviour in another domain. However, only a limited number of studies have experimentally tested this priming hypothesis, and the evidence is mixed. We tested the priming hypothesis in a registered report. We manipulated explorative behaviour in a spatial search task by randomly allocating people to search environments with resources that were either clustered together or dispersedly distributed. We hypothesized that, in a subsequent anagram task, participants who searched in clustered spatial environments would search for words in a more clustered way than participants who searched in the dispersed spatial environments. The pre-registered hypothesis was not supported. An equivalence test showed that the difference between conditions was smaller than the smallest effect size of interest ($d = 0.36$). Out of several exploratory analyses, we found only one inferential result in favour of priming. We discuss implications of these findings for the theory and propose future tests of the hypothesis.

## 1. Introduction

Search and exploration, distinct but intimately related behaviours, are involved in many aspects of human life. We search and

explore when deciding on who to date or marry [1], when navigating information space for answers to our queries [2,3], when testing the outcomes of unfamiliar options [4], when identifying targets in a visual space [5] and when recalling memories or knowledge from our minds [6–9]. It is, therefore, no wonder that scientists have been researching search and exploration behaviour for many years. One of the most interesting claims in the literature is that there may be a domain-general tendency to explore, possibly due to common search processes being involved in different domains (e.g. [10,11]).[1] Perhaps even more intriguing is the finding that experience searching in one domain may influence how people subsequently search in another [12–14]. The test of this priming hypothesis has been interpreted in the literature as an implicit test of the domain-generality hypothesis.

The domains of spatial search, in which people search a spatial environment for rewards, and of cognitive search, in which people search cognitive space for solutions to a problem, have been argued to share similar decision processes, problem structures and neuro-physiological mechanisms [10–14]. For example, when searching a physical environment, people must decide, at any point in time, whether to continue their search where they are or search elsewhere. Likewise, when searching the cognitive environment for solutions, people must decide whether to persist with one line of thinking or switch to another. Hills *et al.* [13] argued that a mechanism originally developed and used to deal with the explore–exploit trade-off in one domain (likely to be spatial search)—the dilemma between exploiting the known and exploring the unknown is typically referred to as the explore–exploit trade-off—could have been exapted through evolution and used in other domains (e.g. internal or cognitive search).

Although domain-specific mechanisms probably exist, Hills and co-workers [6,13] argue that there is evidence to suggest that a domain-general search process may also be at work, and that it is probably a consequence of the same neurological mechanisms that have evolved to control both goal-directed behaviour and goal-directed cognition. For example, dopaminergic mechanisms in humans are involved not only in spatial goal-directed behaviours, but also in goal-directed cognition and general cognitive control, suggesting that such mechanisms are domain-general [6]. Therefore, spatial and cognitive search may have similar physiological underpinnings in the brain.

Hills *et al.* ([12,13], see also [14]) proposed that if, indeed, spatial and cognitive search have a common biological basis and are constrained by similar underlying physiologies, then experience with search in one environment may influence search behaviour in another. Using this logic, they hypothesized, and found, that prior experience with a spatial search environment in which resources were clustered (versus dispersed) caused, or primed, people to subsequently search in a more clustered (versus dispersed) way in a cognitive environment. They thus concluded that search can be primed across domains.

Although the idea of a domain-general mechanism that governs exploration and search is theoretically appealing and has an intuitive adaptational justification, the corresponding cross-domain priming hypothesis still needs further empirical investigation. Given its central importance and implications for the study and modelling of human behaviour (not only in psychology, but in all behavioural, social and computer sciences), the cross-domain priming hypothesis deserves further rigorous testing and to be examined by a multiplicity of different studies. Notwithstanding the breakthrough studies conducted by Hills *et al.* [12,13], the cross-domain priming hypothesis is supported by only a few behavioural studies, and there is also evidence that reports contradictory results, calling for further systematic investigation. There are three compelling reasons for why a large-scale empirical verification of the cross-domain priming hypothesis is needed.

The first reason to further examine the cross-domain priming hypothesis is the possibility that the empirical evidence it relies on is biased in favour of the presence of an effect, such as through publication bias (see, [15–18]). Publication bias occurs where journals, for example, are more likely to publish positive results and statistically significant findings, increasing the likelihood that effect sizes in the literature are inflated. Indeed, the observed effect sizes in Hills *et al.* [12,13] may be inflated. For example, Mekern [19] conducted a close replication of the Hills *et al.* studies, using the same tasks and design, but found a statistically significant effect in the opposing direction: participants who searched for resources in the dispersed (versus clustered) spatial environment subsequently searched in a more clustered (versus dispersed) way in the cognitive environment. However, Mekern's [19] study involved a 6 min task between the spatial foraging and cognitive search tasks. Given the possibility that priming effects of clustered search can dissipate in short time (see [13]), the contradictory findings may be due to

[1]The Stage 2 review process indicated that our wording here may be ambiguous and thus create a somewhat inaccurate representation of the claims in the literature. The past work hypothesizes a domain-general cognitive *search process* that calibrates a tendency towards exploration and exploitation. A consequence of such a domain-general search process is that there is likely to be a domain-general tendency to explore, such that how much people explore and exploit, and how they explore and exploit, will be similar across domains.

methodological differences. Moreover, the manipulation check in Mekern [19] showed that participants in the clustered environments had a smaller average turning angle than those in the dispersed environments, opposite to what was found in previous studies (see [12,13]). However, even though Mekern's [19] manipulation check was in the opposite direction than expected, the effect on the main dependent variable was also in the opposite direction to what was hypothesized, so that it could be argued that their findings are in fact consistent with the priming hypothesis.

To assess the meta-analytic effect size estimate from the three studies (i.e. [12,13,19]) combined, we used Goh et al.'s [20] calculator, inputting the relevant information (reported effect size estimate, group sample sizes, means, standard deviations and/or the t-statistics). We calculated two fixed-effects meta-analytic effect size estimates. First, we calculated the meta-analytic effect size with Mekern's findings interpreted as being in favour of the priming hypothesis, for the reasons provided in the preceding paragraph. Second, we calculated the meta-analytic effect size with Mekern's findings interpreted as being against the priming hypothesis. When Mekern's [19] effect size was input as consistent with the priming hypothesis, the meta-analytic effect size estimate was statistically significantly in favour of the priming hypothesis (mean Cohen's d of 0.64, $CI_{95\%}$[0.32, 0.96]); when input as inconsistent with the priming hypothesis, the meta-analytic effect size estimate was statistically *non-significant* (mean Cohen's d of 0.26, $CI_{95\%}$[−0.06, 0.58]).[2] (See electronic supplementary material for details of parameters obtained from each study for the meta-analysis, and the Open Science Framework, OSF, project page for the calculation of the meta-analytic effect size in the Excel spreadsheet, tables S1 and S2; https://osf.io/fnax6/; check 'READ ME' file first.)

Second, research has shown that support for scientific hypotheses is largely contingent on the subjective choices that inform the design of studies [21], and that research findings are often not generalizable to other stimuli (e.g. [22–26]). Therefore, to examine the robustness, or generalizability, of the hypothesis that search experience in one domain primes search behaviour in another domain, it is necessary to examine the same hypothesis with different design choices. Indeed, Hills et al. [12] noted that further research is needed to identify the extent and boundary conditions of a generalized cognitive search process, using different search tasks.

If there are generalized cognitive search processes at work, then the priming effect described above should not be task-dependent. Although several of the references in the preceding paragraph suggest that researchers should randomly sample stimuli and model them as random factors in their statistical models, such an endeavour is beyond the scope of the present paper. Rather, we aim to examine how well the findings of Hills et al. [12,13] generalize to a different spatial search task. We would expect that experience in a modified multi-armed bandit task (described in more detail below), in which resources are either clustered or dispersed, should prime search behaviour in a subsequent cognitive search task. If we observe a cross-domain priming effect using a different spatial search task, this would provide evidence for the generalizability of the priming hypothesis to at least one other task, increasing confidence in the hypothesis and the theory it supports; if we do not find evidence for such priming, this would indicate that the reliability of the original findings may be more constrained than originally recognized [27].

Finally, there is a third reason to examine the priming hypothesis using a different spatial search task. The findings of Hills et al. [12,13] have an alternative explanation. People in the spatial search task in which the resources were clustered may have been in a heightened state of arousal caused by more frequent key presses, when compared with those who searched the dispersed environments, due to a possible artefact of the task design [12]. Although Hills et al. [12] ruled this out by including the number of key presses as a covariate in an ANOVA, still finding a significant effect in support of the hypothesis, the task we will use in the proposed study addresses this issue by requiring all participants to use the same number of button presses.

Taken together, empirical support for the cross-domain priming hypothesis is still not robust. Therefore, in conjunction with the preceding arguments, it is important not only to examine how well the findings generalize to another task, but also to do so using a registered report format, which removes the problems of publication bias, given that publication is not contingent on finding statistically significant results.

# 2. Overview of the study

We aim to test the hypothesis that experience in a spatial search task will influence search patterns in a subsequent cognitive search task. Specifically, we predict that people who engage in a spatial search task in which the resources are clustered, compared with those for whom the resources are dispersed, will

[2]When the 95% confidence interval crosses over zero, the test is statistically non-significant at the 5% $\alpha$-level.

spend more time in each letter-set of the cognitive search task, as measured by the mean time spent in each letter-set post-test minus mean time spent in each letter-set pre-test. Note, however, that if people in the dispersed environments engage in more clustered search than people in the clustered environments, as will be shown on the manipulation check, then our hypothesis will be reversed such that people in the dispersed condition should spend more time in each letter-set than people in the clustered condition. We will use the same cognitive search task as Hills *et al.* [12,13]: a scrabble-like task in which participants must form words using the letters from a series of randomly generated letter-sets. To examine the robustness and generalizability of the priming hypothesis, we will use a modified multi-armed bandit for the spatial search task. Multi-armed bandit problems are behavioural tasks that have been widely used in research on explore–exploit trade-off decisions (e.g. [28–31]). The vast multi-armed bandit that we propose to use is essentially a large 20 by 20 grid of tiles, with each tile paying either 0 or 100 points once it has been clicked—representing resources distributed in the spatial environment. The tiles worth 100 points will be either clustered together or dispersed across the whole grid, depending on the experimental condition. All participants receive the same number of trials, or button clicks, to search for these resources. Thus, we will test the robustness of the evidence for the priming hypothesis and its generalizability to another spatial search task.

## 3. Pilot test of the spatial (grid) search task

We conducted a pilot test to examine the effect of our proposed manipulation on a manipulation check.[3] We compared participants who were randomly allocated to search clustered or dispersed spatial environments on a measure of clustered search. Details of the task are described in the Methods section.

We aimed for a sample of 100 participants but due to a software error, we lost data from some participants, resulting in a total sample of 86 participants recruited from Prolific.ac. Participants were randomly allocated to engage in the grid search task either with clustered resources ($n = 43$) or dispersed ($n = 43$). We operationalized the clustering of people's search in the grid task by calculating the mean distance of each revealed tile from all other revealed tiles in the grid, taking the average of these mean distances within each grid, and averaging this across all five grids that each participant searched. More technically, for each grid, we collected the $x$ and $y$ coordinates of the 80 selected tiles. For each of the 80 selected tiles, we computed the mean of all Manhattan distances between that tile and all other 79 selected tiles. The Manhattan distance is calculated such that, given two points $(x_1, y_1)$ and $(x_2, y_2)$, the Manhattan distance is $d_M = |x_1 - x_2| + |y_1 - y_2|$, where $|.|$ indicates the absolute value. Now, the 80 means so obtained are averaged, yielding the dispersion measure (or measure of clustered search) of the selected tiles in that grid, and the dispersion/clustering measure of each grid is averaged across all five grids. Thus, for each participant, we had a measure representing the mean distance of each revealed tile from all other revealed tiles in each grid, across all five grids. Higher numbers mean that revealed tiles were further from one another, such that participants searched in a more dispersed manner, and lower numbers mean that participants engaged in more clustered search.

Our pilot test showed that the manipulation had a significant and strong effect on the measure of how clustered/dispersed people searched. Participants in the clustered condition ($M = 10.07$, s.d. $= 2.20$) searched in a more clustered way than participants in the dispersed condition ($M = 12.15$, s.d. $= 2.07$), and this difference was very large, $p < 0.001$, $d = 1.40$. Participants took an average of 7.3 (2.7) min to complete the grid search task: 6.7 (1.9) min for clustered grids and 7.8 (3.3) min for dispersed grids ($p = 0.107$). The data and analysis code can be found on the OSF project page (see link below). We conclude that the manipulation is sufficiently strong for the purposes of the proposed study.

## 4. Methods

The materials for conducting the experiment (i.e. landscapes and letter-sets), R code for data analyses of the pilot data and R code for the main results, an anonymized pilot data file and an anonymized main study data file are on the Open Science Framework (OSF) project page, https://osf.io/qwp84/.

---

[3]A first pilot test used clustered grids with a single large patch of resources but following reviews, we created new clustered grids with multiple patches. The first pilot study also showed that the manipulation was effective: people in the clustered grids ($n = 25$, $M = 6.63$, s.d. $= 1.02$) searched in a more clustered way than participants in the dispersed condition ($n = 31$, $M = 12.35$, s.d. $= 2.02$), and this difference was extremely large, $p < 0.001$, $d = 3.58$.

## 4.1. Participants

We recruited participants from Prolific.co (www.prolific.co), an online participant recruitment tool which provides high-quality data. Participants were required to be fluent in English or have English as their first language. We used the 'small telescopes' approach [32] to determine the required sample size for our study and to evaluate the results. The original study by Hills *et al.* [12] had 20% power to detect an effect size of Cohen's $d = 0.36$, and their replication study [13] had 30% power to detect the same effect size. To achieve 95% power for a one-sided between-subjects $t$-test to detect an effect size of $d = 0.36$, with an $\alpha$-level of 0.05, we required a sample size of 168 per condition (total $N = 336$). Given that our randomization procedure did not allow for perfect control of the number of participants allocated to each condition, and to account for potential data exclusions (described in detail below), we aimed to recruit 400 participants to achieve a minimum of 168 participants per condition for the analyses, after exclusions. To achieve this, we collected 428 observations. Of these, 13 were excluded for not completing the experiment. A total of 19 participants were excluded based on pre-registered exclusion criteria 1–2 (failed either of the comprehension checks more than twice), and 55 participants were excluded based on the pre-registered exclusion criteria 3–7. There were two additional complete responses but for which the software did not register the condition, making these two observations not usable in the analysis. The pre-registered analyses were thus run on a final sample of 339 participants ($n_{\text{clustered}} = 172$, $n_{\text{dispersed}} = 167$); age: $M = 34.6$ (s.d. $= 12.0$); 216 females, and 3 identifying as other.

The study took an average of 28.6 (s.d. $= 10.8$) minutes to complete. Participants were paid a base rate of £2.60 and offered two bonus opportunities (we pre-registered a base payment of £2.00 pounds but, before collecting data, we deemed this insufficient and increased the base payment to £2.60). Participants who receive no inactivity warnings (described below) received a bonus of £0.70, to incentivize the best conditions for priming from the spatial search task to the second anagram task. Participants were also incentivized in the spatial search task with the opportunity to win a £0.70 bonus based on the points they accumulated (described below under instructions to participants).

The target sample size of at least 168 participants per condition, with an $\alpha$ of 0.05, was calculated to give the equivalence test 90% power to reject effects larger than $d = 0.36$. We calculated power for the equivalence test using the TOSTER package in R [33]. Equivalence tests (see [34]) allow researchers to run analyses designed to reject effects at least as large as the smallest effect size of interest. In the present study, if the equivalence test is statistically significant then we can conclude that the true effect size is likely to be smaller than 0.36, and future studies investigating the priming hypothesis should thus be designed with statistical analyses powered to detect effects smaller than this. With a significant equivalence test, we can further conclude that the evidence for the priming hypothesis may be more constrained than originally thought.

## 4.2. Procedure

After reading the consent form and agreeing to participate, all participants read the following instructions:

> In this study, you will first do an anagram task, then a clicking task, then another anagram task. Before starting each of these tasks, you will get instructions on what you need to do and a practice round to become familiar with the task.

> **Important!** Please complete this study without taking any breaks, even in between the different tasks. We have included a variety of different checks to make sure participants are attentive throughout the study.

> If you will complete the task without taking any breaks, you will be paid a **bonus of £0.70**.

> Put differently, if you are inactive for a substantial period of time during or in between any of the tasks, you will receive an inactivity warning and you will not win the bonus.

All participants were then given instructions for how to complete the anagram task followed by three comprehension check questions and a practice session with a single letter-set in which they needed to form two correct anagrams to continue. Following the practice session, participants were given 14 letter-sets in the pre-test anagram task and told that they would need to find a total of 30 words to complete the task. The 14 letter-sets for the pre-test anagram task were the same for all participants, presented in the same order. Next, participants were randomly allocated to either the clustered or dispersed condition and read instructions for the grid search task followed by three comprehension check questions, a practice session with one grid and then eight rounds of the grid search task, each round consisting of a new grid. The grids that participants searched either had clustered or dispersed resources, depending

on condition. Finally, participants completed a post-test anagram task involving a different set of 14 letter-sets. The 14 letter-sets for the post-test anagram task were the same for all participants, presented in the same order. During both the anagram and grid search tasks, participants who were inactive for 50 s were given a warning message and those who received two such warning messages in the grid search task or the second anagram task were to be excluded from the analyses (see exclusion criteria for further details). During the instructions for the anagram and grid search tasks, as well as in the two corresponding comprehension checks, participants who were inactive for more than 150 s were given a warning message and missed out on receiving the bonus for completing the task without breaks.

## 4.3. Tasks and materials

Now we provide the details for each task, including instructions to participants.

*Anagram task.* We constructed 29 letter-sets using the 20 most common letters in English (i.e. K, V, X, Z, J and Q were excluded; as per [12]). We used R to randomly produce letter-sets made up of six letters, four consonants and two vowels, and we continued to generate letter-sets until we had a sufficient number of them with a similar number of average solutions to Hills *et al.*'s [12,13] letter-sets. One letter-set was used for the practice phase, 14 for the pre-test phase prior to the grid search and 14 letter-sets were used for the post-test phase after the grid search. (The letter-sets can be found in the electronic supplementary material: the first letter-set in the list is the practice one, the next 14 are for the pre-test and the final 14 are for the post-test.) Participants were instructed to use any combination of at least four letters from each letter-set to make words. Plurals and proper names were disallowed. Participants were told that the task would end after they found a total of 30 correct words, although they were not told how many letter-sets there were. The letter-sets have an average of 12.7 (s.d. = 4.5) solutions (based on https://wordsmith.org/anagram/advanced.html), which we judged to be close enough to the average solutions per letter-set from the original studies (i.e. $M = 14.7$, s.d. = 5.5; [12,13]). The pre-test letter-sets have a similar number of solutions compared with the post-test letter-sets in the corresponding position (e.g. letter-set 1 in both the pre-test and post-test sessions has 14 solutions). Hills *et al.* [12,13] incorporated a 15 s delay when moving between letter-sets to incentivize people against switching between letter-sets as soon as finding words becomes more difficult, but this will affect participants in both groups equally such that any differences between the conditions must be due to the experimental manipulation. Thus, we saw no reason to include a time cost of switching between letter-sets.

The precise instructions for the pre-test anagram task were as follows:

You will be given an anagram task. In this task, you will be presented with one letter-set at a time, each made up of 4 consonants and 2 vowels. You must use the letters in each letter-set to make up as many English words as you can. You can only use each letter once.

The words you make must have **at least 4 letters** and **cannot be plurals or proper nouns**.

At any time, you can choose to move on to the next letter-set, but you will not be allowed to revisit the previous ones.

The task will end once you have submitted a total of **30 correct words** in total from any number of letter-sets. Immediately after you have entered each word, you will receive feedback about whether it is a correct word.

There are no time constraints for this task, but you should allocate your time appropriately so as not to stay too long or too short in any given letter-set.

Before the task, you will be given one letter-set to get familiar with the task. In this practice round, you will have to submit at least two words to be able to continue with the task.

Following these instructions, participants must answer three comprehension check questions correctly prior to proceeding (after two incorrect attempts, they will be excluded from further participation). The comprehension checks are:

How many letters can the words be made up of?

— at least 3
— at least 4
— at least 5
— I do not know

What other rules apply to the words?

— They can be plurals but not proper nouns
— They can be proper nouns but not plurals

— They cannot be plurals or proper nouns
— I do not know

How many words do I have to form to complete the task?

— 10 words
— 20 words
— 30 words
— I do not know

After correctly answering the comprehension check questions, participants were informed that 'You will now have one practice round before starting the first anagram task. For this practice round, please find and type in **2 words** made up of at least 4 letters from the letter-set below'. The letter-set for the practice round was the same for every participant (i.e. 'Y E A W H S'). After the practice round, participants could 'Click continue to begin the first anagram task'. At the top of the screen, for every letter-set, participants could read the key rules:

Please remember:

— Words must have at least 4 letters;
— Words cannot be plurals or proper nouns;
— The task ends after finding 30 correct words.

After the pre-test anagram task, participants did the grid search task (described below) and then the post-test anagram task. Prior to starting the post-test session, participants were reminded of the main instructions. They were told that:

This anagram task is the same as the first anagram task. The rules are the same.

Please remember:

— task ends after finding 30 correct words;
— must use at least 4 letters;
— no plurals or proper nouns; and
— no time constraints but allocate your time appropriately so as not to stay too long or too short in any given letter-set.

Once participants clicked the 'Continue' button, the post-test anagram task began. Once again, at the top of the screen, for every letter-set, participants could read the key rules of the task (as described for the pre-test session).[4]

**Spatial (grid) search task**. For spatial search, we had participants engage in a variation of the multi-armed bandit task, essentially a grid search task. For each category of environment type (clustered versus dispersed resources), we generated forty 20 by 20 grids (with a total of 400 tiles in each grid) using the R packages *NLMR* and *landscapeR* for simulating landscape models [35,36]. Clustered landscapes were generated using the makeClass function, which simulates a patchy landscape model, in which the number of patches ranged from 3 to 5 and was randomly determined. The size of each patch ranged from 10 to 20 and was also randomly determined (independently from the number of patches). We iteratively created patchy landscapes with these parameters, and retained those that had a total number of reward tiles between 77 and 83. Dispersed landscapes were created by using the nlm_neigh function, which implements the neighbouring method by Scherer *et al.* [37]: in our landscapes, given a target tile with a positive payoff, we set to 0.001 the probability of placing another tile with a positive payoff in the Moore-neighbourhood of the target tile (i.e. in one of the eight cells surrounding it). Details of the procedures and the R code for generating the landscapes, as well as the images for the landscapes that we created, can all be downloaded and viewed from the OSF project page. The 40 grids for each of the two environment types had extremely similar average points per

---

[4]During Stage 2 reviews, we realized that some additional details of the anagram task would be beneficial. Specifically, during the pre-test and post-test anagram tasks, participants were shown the number of correct words in total they had formed thus far in the task and a list of the correct words they had formed from the current letter-set.

(a)    (b)

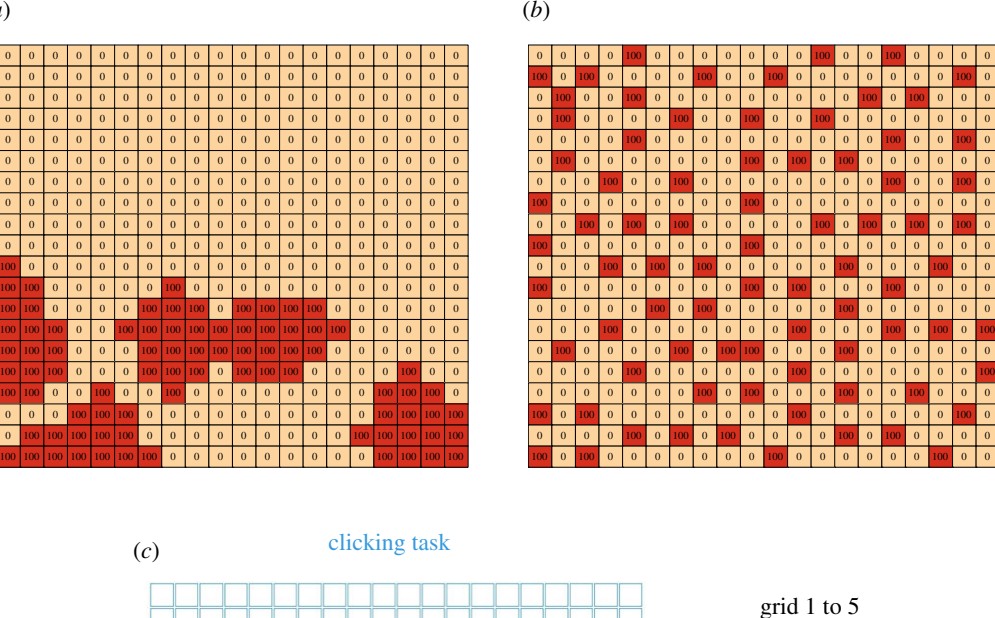

(c)    clicking task

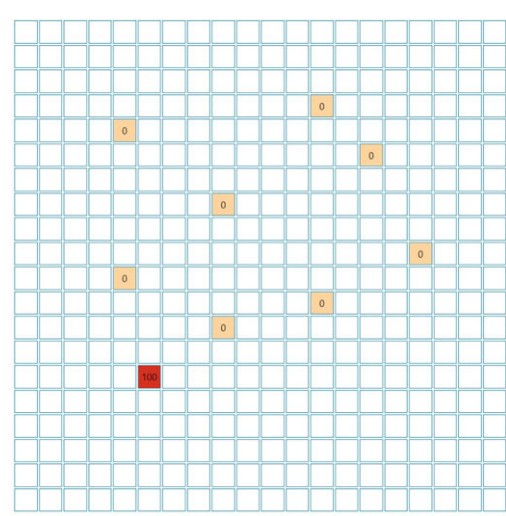

grid 1 to 5

remaining clicks for this grid: 71

points from last click: 100

points cumulated in this grid: 100

**Figure 1.** (a) An example of a clustered search environment, whereas (b) that of a dispersed search environment. (c) How the task appears to a participant after nine tile selections.

tile: in the clustered landscapes, the average points per tile is $M_{clustered} = 20.13$ (s.d._clusterd = 0.49) and, as noted earlier, there are between 77 and 83 reward tiles in each landscape; in the dispersed landscapes, the average points per tile is $M_{dispersed} = 20$ (s.d._dispersed = 0) and there are exactly 80 reward tiles in each landscape. The difference depends upon the algorithm used to populate the landscapes. Figure 1 presents examples of clustered and dispersed environments.

The clustered grids consist of resources distributed in three to five patches. The dispersed grids consist of resources distributed apart from each other. Therefore, as in the original studies, the clustered landscapes involve multiple patches of resources, so that participants can switch between exploration and exploitation within each landscape. Participants engaged in the grid search task over five rounds, each on a different grid randomly drawn (without replacement) from the 40 available grids of the relevant environment type. In each round, participants had 80 trials in which they could select any tile and gather points (or resources) which they accumulated. Participants' task was to maximize the number of points they accumulated to increase their chances of winning a bonus payment in addition to their base payment. Participants could click any of the tiles in the grid (without re-clicking previously revealed tiles) until they exhausted their 80 trials. Clicking unrevealed tiles would display whether any resources were associated with that space and participants either gained 0 or 100 points (see figure 1c). Tiles could not be re-clicked.

The precise instructions to participants for the spatial search task were as follows:

In the following task, you will be presented with a series of **5 different grids** to explore. By clicking on tiles in the grid, you reveal points that are associated to the location on the grid. On each grid, you will have **80 clicks**, with

the number of remaining clicks displayed on the side of the grid. When you run out of clicks, you will start a new trial on the next unexplored grid.

Use your mouse to click and reveal new tiles, which will display a number corresponding to the number of points you gain. The points that you can earn in each tile are either 0 or 100. Revealed tiles are also colour coded, as a visual aid to help you in this task. Darker colours correspond to larger rewards. **Each tile can be selected only 1 time.**[5]

Before the task, you will be given one additional grid to click on to get familiar with the task. In this practice round, you will also have 80 clicks.

**Important!** It is your task to gain as many points as possible across all 5 grids. The total number of points you accumulate across all 5 grids will increase your chances of winning a **bonus of £0.70**. The average points per click will give the chance of winning the bonus: For example, if on average you have earned 50 points per click, then you will have a 50% chance of winning the bonus.

Following the instructions, participants were given three comprehension check questions which needed to be answered correctly prior to proceeding (after two incorrect attempts, they were excluded from further participation). The comprehension check questions were as follows:

What is your task in this study?

— Counting tiles
— Gain as many points as possible
— Learn where the largest reward is
— I do not know

How many grids are there to explore?

— 1 grid
— 4 grids
— 5 grids
— I do not know

How many times can you click on the same tile?

— 1 time
— 8 times
— unlimited times
— I do not know

After the comprehension checks, participants engaged in a practice phase of 80 trials in a single grid to become familiar with the task. This practice phase is not included in the analyses. After the practice phase, participants were told that 'Once you click continue you will begin exploring the first out of the 5 grids for this task', after which the grid search task began, consisting of 80 trials in each of five grids. The number of points accumulated and the number of remaining clicks were both displayed to the right of the grid. After the fifth and final round of the grid search task, participants were given a summary of their performance and told their chances of winning the bonus (e.g. if they had won an average of 50 points per click, then they had a 50% chance of winning the bonus payment). Participants were informed about whether they won the bonus at the end of the experiment. Clicking 'Continue' took participants to the post-test anagram task.

One benefit of using the multi-armed bandit is that all participants had a fixed number of trials. Hills *et al.* [12] noted that an alternative explanation for their results is that participants in the clumpy environment may have been in a heightened state of arousal due to more frequent clicking of response keys. To address this issue, they included the number of key presses as a covariate in an ANOVA and still found the significant effect of the manipulation. However, given that the number of key presses is a proxy measure of arousal, there is always the potential for residual confounding. In other words, controlling for the number of key presses may not perfectly control for the state of arousal across the conditions. In the grid search task we have described, all participants across all conditions had only 80 clicks per round, resolving the issue of differences in states of arousal due to the frequency of key presses.

---

[5]In the pre-registered manuscript, the instruction to participants said, 'previously revealed tiles cannot be reselected'. However, to improve clarity, prior to data collection, we changed this to 'Each tile can be selected only 1 time'.

## 4.4. Measures

The measure of clustered search, or manipulation check, were calculated as done in the pilot study described above.

The dependent variable of interest is the difference in the mean time spent in each letter-set going from the pre-test anagram task to the post-test anagram task. For each participant, we calculated the mean time spent in each letter-set in the pre-test anagram task ($t_{A1}$) and the mean time spent in each letter-set in the post-test anagram task ($t_{A2}$). In both cases, we excluded the final letter-set in which the 30th word was found because participants would be likely to have stayed in the final letter-set for a shorter time, given that the task was likely to end prior to them having exhausted their search in the letter-set. We calculated the difference $\Delta_t$ in the mean time spent in each letter-set by subtracting $t_{A1}$ from $t_{A2}$ (i.e. $\Delta_t = t_{A2} - t_{A1}$). The difference $\Delta_t$ was the dependent variable for our analyses (as per [12]).

# 5. Analysis plan

First, we will examine the effect of the manipulation on the manipulation check. Specifically, as we did in the pilot study, we will examine whether people in the clustered search environments explored the grids in a more clustered way than people in the dispersed search environments.

## 5.1. Exclusion/inclusion criteria

The exclusion/inclusion criteria are as follows:[6]

(1) participants who answer the comprehension check questions for the anagram task incorrectly more than two times will be excluded from the study and the analyses;
(2) participants who answer the comprehension check questions for the grid search task incorrectly more than two times will be excluded from the study and the analyses;
(3) participants who do not find a minimum of 20 words in each of the anagram tasks will be excluded from the analyses (i.e. only those who find at least 20 words in both the pre-test and the post-test sessions will be included in the analyses);
(4) participants who get two (50 s) inactivity warnings in the post-test anagram task will be excluded from the study and the analyses;
(5) participants who get two (50 s) inactivity warnings in the grid search task will be excluded from the study and the analyses;
(6) participants who are given an inactivity warning and who are inactive for 100 s or more during that warning will be excluded from the analyses; and
(7) participants who spend more than 150 s on the instructions for the post-test anagram task will be excluded from the analyses (to avoid the possibility for a priming effect to wear out).

Although participants who get a 150 s inactivity warning in the instructions for the grid search task will miss out on the bonus intended for completing the study without breaks, they will not be excluded from the analyses.

## 5.2. Pre-registered analyses

Our hypothesis was that the distribution of resources during search in a spatial environment will prime how people search in a cognitive environment. Specifically, people in the grid search task with clustered resources, compared with people searching for dispersed resources, would spend more time in each letter-set of the post-test anagram task, as measured by the mean time spent in each letter-set post-test minus the mean time spent in each letter-set pre-test. That is, the difference in the mean time spent in

---

[6]We first pre-registered the manuscript with criterion 3 requiring participants to find a total of 30 words to complete the task and to be included in the analyses (see pre-registration here: https://osf.io/2gu46). But before collecting data, we contacted the editor who approved the change of this criterion so that participants needed to find 20 words in each of the pre-test and post-test anagram tasks to be included in the analyses (i.e. they would be excluded if they did not achieve this minimum). The updated pre-registration said that 'Participants who do not complete the anagram tasks will be excluded from the analyses (i.e. only those who find 20 words in both the pre-test and the post-test sessions will be included in the analyses)' (pre-registration here: https://osf.io/jge9p). But to complete the task, participants needed to find 30 words and to be included in the analyses they needed to find 20 words. We have therefore changed this in the manuscript to reflect the fact that participants who did not find the minimum of 20 words were excluded from the analyses, as opposed to those who did not complete the task.

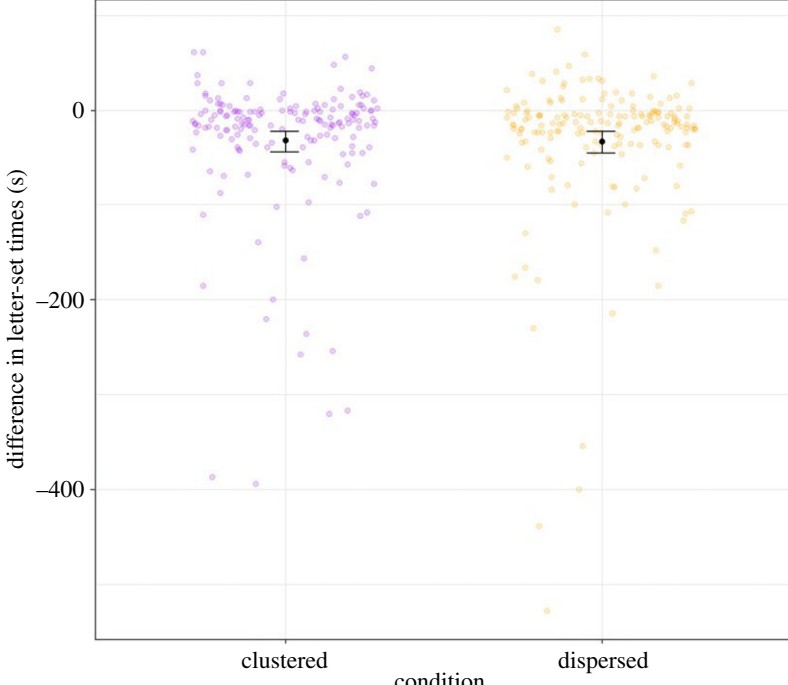

**Figure 2.** Black dots and whiskers reflect the means and the 95% confidence intervals around the means, respectively. The y-axis is the outcome variable (i.e. difference in the average time spent in each letter-set between the post-test anagram task and the pre-test anagram task, $\Delta_t = t_{A2} - t_{A1}$, where $t_{A2}$ is the average time in each letter-set in the post-test anagram task and $t_{A1}$ is for the pre-test anagram task). The coloured dots represent participants based on their score on the outcome variable.

each letter-set between post-test and pre-test would be greater for people who searched spatial environments with clustered resources than those who searched dispersed environments. Referring to the notation introduced earlier, this is equivalent to hypothesizing that $\Delta_t$ (as defined earlier) for those in the clustered condition would be greater than $\Delta_t$ for those in the dispersed condition. To test this hypothesis, we used Welch's t-test against 0 (which does not assume homogeneity of variance), comparing people in the clustered and dispersed conditions on the difference in mean time spent in each letter-set.

Finally, we conducted an equivalence test using the TOSTER package, with equivalence bounds of $d = -0.36$ to $d = 0.36$. If the equivalence test is statistically significant, we can conclude that the true effect is likely to be smaller than 0.36, that future tests should thus have appropriate statistical power to detect such effects, and that the priming hypothesis should be further examined to identify more precisely how constrained it is.

## 6. Results

The manipulation check showed that the manipulation was successful. People in the clustered condition ($M = 9.95$, s.d. $= 1.34$) searched in a more clustered way than participants in the dispersed condition ($M = 12.07$, s.d. $= 1.71$), and this difference was very large, $t_{313.90} = 12.71$, $p < 0.001$, $d = 1.38$ CI$_{95\%}$[1.15, 1.62].

### 6.1. Results from pre-registered analyses

The hypothesis was not supported. The difference in the mean time (in seconds) spent in each letter-set, from pre-test to post-test (from now on called the outcome variable), with negative numbers representing less time spent in each letter-set post-test, was not statistically different between the clustered condition ($M = -33.22$, s.d. $= 77.86$) and the dispersed condition ($M = -32.12$, s.d. $= 72.22$), $t_{336.30} = 0.14$, $p = 0.554$, $d = -0.02$ CI$_{95\%}$[−0.23, 0.20]. Moreover, the equivalence test was significant, $t_{336.30} = 3.18$, $p < 0.001$, showing that the observed effect was statistically smaller than the equivalence bounds. Hence, the observed difference between the clustered and dispersed conditions was smaller than the smallest effect size of interest (i.e. $d = 0.36$). The results are visually represented in figure 2.

## 6.2. Results from (non-pre-registered) exploratory analyses

A reviewer suggested we use an alternative manipulation check in which clustered search is examined in a sequential step fashion. That is, for each selection at time $t$, the distance from the previous selection at time $t - 1$ provides a measure of how distant/close that selection was. And the average of this distance across all trials in a block provides a measure of clustered search. Using this outcome as a manipulation check, we also find that people in the clustered condition ($M = 2.57$, s.d. $= 0.54$) searched in a more clustered way than people in the dispersed condition ($M = 4.27$, s.d. $= 1.66$), and this was also a large effect, $t_{199.92} = 12.60$, $p < 0.001$, $d = 1.38$ CI$_{95\%}$[1.14, 1.61]. Thus, regardless of which manipulation check we used, we can conclude that the different environment types produced different search styles with the clustered environments resulting in more clustered search than the dispersed environments.

A reviewer suggested that a priming effect may depend on resource-dependent shifting between local and global search. That is, priming may occur only when the different environment types make people change their search strategies depending on whether they find a resource or note. Specifically, people in the clustered environments should search more locally after finding a resource, compared with after not finding a resource, to increase their chances of finding another resource tile nearby. By contrast, people in the dispersed environments should search more globally after finding a resource, when compared with after not finding a resource, to move away from the last found resource rather than staying local where resources would not be expected. This was found in [12] (see their fig. 2) in terms of turning angle, and in Wilke *et al.* [38] in terms of distance jumped in a grid search task (where win-stay versus win-shift behaviour was seen).

To test this, we used the sequential step clustered search measure described in the preceding paragraph to calculate the average search distance on trials after a resource was found and compared this with the average search distance on trials after a resource was *not* found. We did this separately for the dispersed and the clustered conditions. For the dispersed condition, the average distance after finding a resource tile ($M = 4.54$, s.d. $= 2.07$) was slighter larger than after not finding a resource ($M = 4.29$, s.d. $= 1.70$), $t_{166} = 2.15$, $p = 0.033$, $d = 0.12$ CI$_{95\%}$[0.01, 0.24]. By contrast, for the clustered condition, the average distance after finding a resource ($M = 1.49$, s.d. $= 0.47$) was much smaller than after not finding a resource ($M = 4.25$, s.d. $= 1.04$), $t_{170} = 36.35$, $p < 0.0001$, $d = 3.36$ CI$_{95\%}$[2.89, 3.83]—note that the degrees of freedom are correct here because there were 171 observations for this measure since one subject had a very unlucky situation in the clustered condition in which they found zero rewards in one of the environments. Therefore, we observed the expected resource-dependent shifting between local and global search in both the clustered and dispersed environments.

We conducted exploratory analyses to examine whether there was an effect of priming in the earlier letter-sets of the post-test anagram task that might have worn out with time and thus been hidden when we analysed the time averaged across all of the post-test letter-sets. To do this, we calculated the difference between the average time spent in each letter-set pre-test and the time spent in the first letter-set post-test. Then we did the same using only the time spent in the second letter-set post-test, then the third letter-set post-test, and so on until the sixth letter-sets. And we compared the clustered and dispersed groups for each of these dependent variables. None of these tests showed a statistically significant difference between the clustered and dispersed groups (all $p$s $> 0.33$).

We then compared the time spent in each type of landscape in the grid search task (excluding the practice trial). Overall, participants spent an average of 6.42 (s.d. $= 4.93$) min in the grid search task. People in the clustered condition spent fewer minutes in these landscapes ($M = 5.57$, s.d. $= 1.62$) than people in the dispersed condition ($M = 7.30$, s.d. $= 6.73$), $t_{184.54} = 3.22$, $p = 0.0015$, $d = 0.35$ CI$_{95\%}$[0.14, 0.57]. To test whether the average time spent in the landscapes was related to the outcome variable, we ran a regression analysis. Time spent in landscapes was not related to the outcome variable, $b = -0.006$, $p = 0.655$. Moreover, including landscape time together with condition as predictors of the outcome variable did not change the results: condition still had a non-significant effect on the outcome variable, $b = -0.002$, $p = 0.829$.

We tested whether the average time spent in each letter-set pre-test and post-test was related to clustered search (i.e. the manipulation check) in the grid search task, as a way of examining potential individual differences in how clustered people searched across the two domains. If individual differences exist, then there should be a negative correlation such that the longer people spend in each letter-set (i.e. the more clustered they searched in the anagram task), the smaller the distance between the selected tiles in the grid search task (i.e. the more clustered they searched in the grid search task). Our pre-registered manipulation check was not significantly correlated with the average time spent in each letter-set either in the pre-test anagram task ($r = 0.04$, CI$_{95\%}$[−0.07, 0.14], $p = 0.497$) or the

post-test anagram task ($r = 0.08$, CI$_{95\%}$[−0.02, 0.19], $p = 0.146$). Similarly, the reviewer-suggested manipulation check, as described above, was not significantly correlated with the letter-set times in the pre-test anagram task ($r = 0.06$, CI$_{95\%}$[−0.05, 0.16], $p = 0.289$) or the post-test anagram task ($r = 0.07$, CI$_{95\%}$[−0.04, 0.17], $p = 0.216$).

We examined whether the number of resources found, or points earned, in the grid search task was related to the outcome variable. Using a regression, we found that the relationship between points participants earned and the outcome variable was not significant, $b = 0.03$, $p = 0.955$. Nevertheless, people in the clustered condition found on average more resources ($M = 27222.67$, s.d. $= 3516.93$) than people in the dispersed condition ($M = 10144.91$, s.d. $= 1329.63$), $t_{220.14} = 59.46$, $p < 0.001$, $d = 6.42$ CI$_{95\%}$ [5.89, 6.95]. We therefore ran a regression analysis to examine the effect of condition on the outcome variable while controlling for the number of resources found in the landscapes. The results did not change, as the effect of condition on the outcome variable remained non-significant, $b = −0.0002$, $p = 0.527$.

Finally, we examined whether there was a priming effect within letter-sets, based on the words participants formed (in line with [13]). To this end, we computed the bigram similarity between the words submitted consecutively, which reflects how similar (or distant) each formed word is in relation to the previously formed word, with higher scores reflecting greater similarity. Given two words, the bigram similarity was computed as the number of bigrams shared between them (including the 'start-Letter' and 'Letter-end' bigrams; cf. [13]). For example, the word 'tale' is formed by the bigrams: 'start-t', 'ta', 'al', 'le' and 'e-end', and its bigram similarity with the word 'gale' is, therefore, 3 (sharing the latter three bigrams of 'tale'). We tested the effect of the manipulation on bigram similarity of words formed in the post-test anagram task using a $t$-test and found that the people in the clustered condition formed words with lower bigram similarity ($M = 1.27$, s.d. $= 0.23$) than people in the dispersed condition ($M = 1.32$, s.d. $= 0.24$), $t_{334.71} = 2.14$, $p = 0.033$, $d = 0.23$ CI$_{95\%}$[0.02, 0.45]. Examining this relationship in a regression analysis with bigram similarity in the pre-test anagram task entered as a covariate did not qualitatively change the inferential results ($b = −0.05$, $p = 0.046$). Therefore, it may be that the dispersed environments, compared with the clustered environments, led people to produce word solutions with higher similarity to the words preceding each solution.

Although the direction of these results is in the opposite direction to the priming hypothesis, it is in line with the findings of Hills et al. [13], where it was argued that people in the clustered condition may have been primed to persevere for longer on each solution they found. In this way, they may 'stay in the current working memory patch longer' [13, p. 18], making alterations to that one solution repeatedly, producing several subsequent solutions but where each of these would be more distant from its preceding solution. By contrast, people in the dispersed condition may have given up on making alterations to each solution as soon as a new solution arose so that each subsequent solution would be more similar to the preceding one.

# 7. Discussion

This study tested the priming hypothesis stating that exploration in a spatial domain will influence how people explore in a cognitive domain. Past research has found that experience with searching spatial environments in which the resources are either clustered together or dispersed apart will prime people to search for words in an anagram task (i.e. a cognitive environment) in either a clustered or more dispersed manner [12,13]. To examine how far the evidence for the priming hypothesis generalizes, we tested it using a different spatial search task but the same anagram task. Our main pre-registered hypothesis was that people in the spatial search task in which the resources were clustered together, when compared with those in which the resources were dispersed, would spend more time in each letter-set of the post-test anagram task (relative to the pre-test anagram task). We found no support for this hypothesis. In fact, the equivalence tests showed that the effect size we observed was smaller than the smallest effect size of interest, as determined by the small telescopes approach on the findings of past research.

There are at least three possible explanations for our results. First, and perhaps the most parsimonious, is that exploration in one domain does not prime exploration in another domain. This explanation has several implications. It could be that, contrary to past theorizing (e.g. [10–14]), there is no domain-general cognitive search process and, consequently, no domain-general tendency to explore. That is, the cognitive search process may not be a single entity that is domain general, but rather a host of subprocesses that operate during different tasks. Therefore, there is no one thing to be primed. But this conclusion would be too strong given the evidence. It may be that there is a domain-

general search process but that priming does not occur; the results in the published literature potentially being due to publication bias. That is, search processes may be shared across domains, but they are not able to be primed. An alternative implication is that search processes across tasks may be shared but they rapidly adapt to new task demands so that priming does not occur. At this stage, all such conclusions would be somewhat premature. All we can really say is that in our study priming did not occur at a sufficient level to be detected by the analyses. Importantly, the point estimate of the observed effect size seems very closely centred on zero ($d = -0.02$), though the confidence intervals did include effect sizes ($CI_{95\%}[-0.23, 0.19]$) that, while small by Cohen's arbitrary thresholds, may still be considered theoretically relevant.

A second possible explanation is that the priming effect, as observed by Hills et al. [12,13], may be more constrained than originally thought. That is, there may be boundary conditions that we were yet unaware of. Indeed, it may be that the spatial search task we used was too different from the original search task. For example, in the original task used by Hills et al., people could move between patches but it was costly in terms of effort and time because it took time to move across the spatial environment. Therefore, the costs of searching locally were lower than the costs of searching globally. By contrast, in our task, the costs of local and global search were virtually equivalent since people could jump from one side of the grid to the other from one click to the next. Future research and theorizing should examine the features of the spatial search task used by Hills et al. and determine which are necessary for priming to occur. Our spatial search task shared many of the important features with the original task: (i) resources were either clustered together or dispersed; (ii) there were multiple patches of resources in the clustered condition; and (iii) resources, and patches, could be depleted (i.e. each tile could be selected only once). If there are other features of the original task that are necessary for the priming effect to occur, then this should be theorized and hypothesized, with future research addressing these systematically. In this way, we can determine how broad and generalizable the priming effect is.

Indeed, future researchers can test the priming hypothesis using a spatial search task a little more similar to the original studies than our task was and see if the hypothesis holds. The tasks being tested in this fashion could then iteratively be changed to becoming more and more similar to the original task until we identify the boundary conditions of the priming effect, or until there is a very close replication of the original studies, using the same task. Such an iterative process of replication was outlined by Nosek & Errington [27].

A third possible explanation for the results is that there may need to be an extremely strong manipulation for priming to occur. We think this is the least likely explanation and probably the most difficult to test, given that this reasoning could be applied to all null and negative results until a positive result is found.

In addition to the pre-registered hypothesis test, we ran exploratory analyses to examine other factors that might have influenced the results or that might provide some evidence of priming. For example, priming may wear out relatively quickly, such that a priming effect might only be observed for the first few letter-sets in the post-test anagram task (e.g. [13]). However, we found no statistically significant differences between groups on the outcome variable when we considered only the first, second, third, fourth, fifth or sixth letter-sets of the pre- and post-anagram tasks. In addition, although participants in the clustered group finished each grid faster and found more resources (or earned more points) than those in the dispersed group, including the time in each grid or the number of resources found as covariates in regression analyses did not change the inferential results: condition still had non-significant effects on the outcome variable. We also tested whether how clustered or dispersed people searched in the spatial environment (i.e. the outcome variable) was related to how clustered or dispersed people searched in the anagram task (i.e. how much time they spent in each letter-set). However, neither of the two clustered search measures for the spatial search were related to the letter-set times in the pre-test or the post-test anagram tasks.

And finally, we followed the procedure of Hills et al. [13] to examine whether experience in the spatial search task influenced the similarity of word solutions in each letter-set. Essentially, this analysis examines the clustering of search within letter-sets, as opposed to across letter-sets. According to Hills et al. [13], people in the clustered condition may be primed to persevere for longer on the process of iteratively manipulating a letter-set or solution to find new solutions. If this occurs, we should observe people in the clustered condition, when compared with those in the dispersed condition, producing words with less similarity to the preceding word. This is because they would be making more manipulations to the earlier solutions, as opposed to the immediately preceding solution. Our analysis supports this reasoning. People in the clustered group produced words that were statistically

significantly less similar to the immediately preceding word, when compared with people in the dispersed group.

Notwithstanding the exploratory finding and the explanation described in the preceding paragraph, and that these results are in line with the findings reported by Hills *et al*. [13], we recommend some caution in drawing strong conclusions here. The main test of the priming hypothesis was not supported. Out of several different exploratory tests examining whether priming occurred, we found only one that was statistically significant. Given the many exploratory inferential tests that we conducted, the type I error rate (i.e. the probability of finding a statistically significant effect when there is no true effect) is likely to be inflated. We recommend that future research be conducted to examine the priming effect both within and across letter-sets either in a pre-registered study or, ideally, as a registered report, where the type I error rate is controlled by adjusting the $\alpha$-level for the critical tests of the hypothesis. In this way, combined with the iterative process of replication we described earlier, we can be more confident that the result is robust and in identifying boundary conditions for the priming effect.

Ethics. Protocols for all studies were approved by the Research Ethics Committee of the University of Southern Denmark, approval number 19/71830.
Data accessibility. Data and analysis code for the pilot study, as well as the data and analysis code for the main study, can be found at https://osf.io/qwp84/ (check 'READ ME' file first). The manuscript was pre-registered on the OSF: https://osf.io/jge9p. We report how we determined our sample size, all data exclusions, all manipulations and all measures.

The data are provided in electronic supplementary material [39].
Authors' contributions. Both authors conceptualized the study. The first author, F.A., wrote the first draft and both authors contributed to editing the draft. The second author, D.M., wrote the analysis script and programming for the study.
Competing interests. We declare we have no competing interests.
Funding. This work has been funded by a grant from the Danish Council for Independent Research (D.M., DFF—7015-00050). The funding source had no involvement in the study design; data collection; analysis and interpretation of data; in the writing of the report; or in the decision to submit the article for publication.

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
