## [Peer Review File · Royal Society Open Science]

Review History

RSOS-201944.R0 (Original submission)

Review form: Reviewer 1

Do you have any ethical concerns with this paper?

No

Recommendation?

Accept in principle

Comments to the Author(s)

The proposed research aims to replicate research of my own, with some improvements, focused around the question of whether or not there is a domain general cognitive search process that can be primed from one task to another. Naturally, having done the work myself once some time ago, I think the question is still interesting and the replication is important. The authors do an

excellent job of describing what they plan to do and why, and I think barring a few minor corrections, it sounds great.

The Mekern et al study cited within as an example of a failure to replicate is really good work. I communicated with Vera Mekern and Bernhard Hommel about the work and some aspects of the design as well as the results. Several differences in the study are quite important, and I think, sadly, confuse the validity of the effect size comparison. First, the Eye-blink rate task is a 6-minute eye fixation task that takes place before and after the spatial foraging task. In Hills et al. 2010, we show that the priming effects we observed only lasts for about 3-minutes, mostly for the first two letter sets. So I wouldn't have expected the priming to survive the 6-minute EBR. Second, participants in Mekern turn more (with a higher turning angle) in the diffuse environment, which isn't what we observed in previous work and indeed would have likely been impossible to manage accurately in the 2008/10 work because the cursor was moving too quickly. I can't explain that difference, though Mekern suggests it may have had to do with the focused-meditation-like nature of the EBR. I really don't know. Possibly that led individuals to persevere on tiny patches in the diffuse task so they then primed themselves to persevere more in the scrabble task. Third, because of this flip in the spatial foraging task results, one could make flipped predictions about what should happen in the scrabble task as well, and of course that isn't helpful, not to mention the 6-minute EBR should have evened things out anyway. So that's all fiddly and complicated and I don't know what to make of it. One thing Vera's excellent work does show, however, is that the EBR is unaffected by the foraging perseveration in the spatial foraging task that precedes it. In this case, it doesn't matter how the spatial foraging task goes, as long as the conditions are different, then one would predict a difference in dopaminergic processes underlying search, if they indeed underly this search. The failure to find a difference in the EBR after the spatial foraging suggests this doesn't work via that mechanism. I am not an expert in EBR, but I trust that Hommel is, so I am inclined to believe that the mechanisms driving our results in 2008/10 may not work as previously proposed.

Back to the newly proposed research: There are a few ways in which the proposed paradigm diverges from the original. These are not necessarily criticisms and I think at least one is an improvement and necessary. However, a couple deserve a bit more thought.

1. Participants will be exposed to 14 letter sets before the spatial foraging task. In the original research (Hills, 2008,2010), this was substantially fewer in the pre-test. Arguably, 14 before and after is better for comparison, assuming that something unforeseen doesn't otherwise influence behavior (such as boredom or learning). There is no reason to expect these unforeseen things, so I'm inclined to think this is in an improvement.
2. Participants are able to submit three-letter words. The original uses four-letter words. I worry that three-letter words will allow participants to find solutions with less effort. This is a problem if one believes that the effortful part of the task is the part being primed, which I do. I vaguely recall spending some time on finding letter sets that provided enough four-letter solutions. Should this really matter? I'm not sure. In semantic memory search tasks, like category fluency, there is evidence of an executive search effect (e.g., Rosen & Engle, 1997). One would only expect to see this effect during effortful components of a search. If the search task is too easy, then it's not clear what to expect.
3. The proposed paradigm uses a 0-second time cost between letter sets. The original used a 15-second waiting period. The reason for this in the original research is because the task is designed to represent a patch-foraging task, similar to the spatial foraging task, and also similar to the marginal value theorem. According to the marginal value theorem, if there is no travel time between patches, and patch resource intake rates are always monotonically declining after arrival, then foragers should move between patches instantly. Obviously that's in the

mathematical limit of no-travel costs whatsoever, which is unlikely to exactly match the anagrams. It takes time to read the letters and start thinking about how they could be manipulated. However, a 0-second waiting time invites participants to give up as soon as they encounter any difficulty, as the next patch will be a rich source of new solutions. Alongside the 3-letter words, I think this is likely to make the task too easy. Of course, it may not, but I feel it's better for me to point this out even if I'm wrong about how the task will work in practice. I guess I would ask how long it takes participants to find 30 items and how quickly do they switch between letter sets? If this comparable to the earlier work, then it is a better replication.

4. The grid search is the biggest difference from the prior work. I think this is an improvement. As the authors note, this tests not so much a replication as a test of the generalizability of the results to a very similar situation. It also potentially overcomes the arousal problem. Moreover, it avoids the problem observed in Mekern, that the spatial foraging manipulation didn't replicate.

5. In the analysis, will the analysis include a look within individual letter sets, as done in 2010?

6. Also, will the analysis include a focus on individual differences between the two tasks, following the last paragraph of the results in Hills et al., 2008? Neither 5. or 6. were done in Mekern's work, but I think they would have added value. Even if the manipulation failed to work as expected, the replication of a relationship between foraging externally and internally was still available for testing.

Overall, I think the design is a close replication with some minor variations and would be interesting and worthwhile. Plus the authors provide confidence that the work will be well done. The authors should consider whether they think the comments above warrant changing the design to limit solutions to only four or more letters and to increase the wait between new letter sets, as well as whether the final two comments (5 and 6) would further help address the underlying research question about a domain general search process.

Review form: Reviewer 2

Do you have any ethical concerns with this paper?

No

Recommendation?

Accept with minor revision

Comments to the Author(s)

Search for resources of different types is a crucial adaptive behavior for all mobile organisms, and there is ongoing debate about whether identical, similar, or different cognitive mechanisms are used for search in different domains. Priming studies between different search tasks have been used to inform this debate, but the results have been mixed, so a high-powered pre-registered replication would go far to advance this discussion and the understanding of search cognition. This proposed study design looks to be a good way to replicate and test the generality of earlier search priming study results, including of Hills, Todd, and Goldstone (2008, 2010) and Mekern (2017). The authors make good arguments for the importance of replicating the earlier studies. Their new design decisions are also appropriate, including increasing the power with a larger sample of participants, using a number of exclusion tests to ensure consistent sustained attention of the online participants, and having a fixed number of possible search actions per landscape to remove one possible alternative explanation of search priming results in terms of key-press based differences in arousal between the conditions. The methods are described in a clear and detailed

manner that will allow others to readily replicate this study. The results of this study will be very useful in assessing the evidence for priming between different search domains, and consequently for pushing the field of search cognition forward.

There are though a few differences in this design from past study designs that should be considered if the results of this replication study are inconclusive. These are not arguments for changing the proposed design (aside from adding some additional measures and analyses, mentioned below in the detailed comments by page), but rather some further ideas about what could lead to priming between search tasks, if/when it does exist.

First, in the Hills et al. and Mekern spatial navigation search, and in the anagram task, it takes longer to make "bigger" steps (over the landscape, or to a very different word), but the current clicking search task is more like visual search, where longer steps across the landscape do not take (much) longer to produce (participants just have to move the mouse). Does that difference in the spatial task versions break down the connections between the two search tasks in some way?

Second, the fact that the current landscapes only have one big patch means there is no "exploit-to-explore" (patch-leaving) decisions to be made, which is different from the previous versions where there were multiple patches and so multiple switch-patch decisions. The anagram word puzzle includes multiple exploit-to-explore decisions, and in fact that is effectively the main thing being measured. How will the single patch landscapes change priming and behavior? (An alternative design would be to still have one patch, but of varying size across different landscapes, and let participants search as long as they want and switch to the next landscape whenever they want – this would invoke the exploit-to-explore decisions in the Hills et al. and Mekern designs, and in the anagram task.)

Third, because participants in the clustered landscape of the current design only explore one big patch continuously (once they find it) and get a near-constant stream of reward, they might therefore think the world is richer overall than do participants in the diffuse environment, because they have seen a much higher rate of return while searching. Would that different estimate of the "richness" of the world between the patchy and diffuse conditions affect behavior in the anagram task? If so, that could possibly mask any effect of the structure of the environment (clustered vs. diffuse). It could be that participants in the clustered environment would be primed to think the world is very rich, e.g. full of big patches, which could lead them to then stay longer in each patch in the anagram task. Or possibly if they are primed to think the world is rich, then when they are in each anagram patch and they feel themselves slowing down in producing more words, they may think, "this *isn't* one of those big rich patches, so I should leave it right away and look for a big patch!" --leading to the opposite behavior, namely leaving sooner. Either way, if the single-cluster manipulation in the proposed study changes how rich the spatial world feels, that could make a difference in the results for an unintended reason.

Detailed comments by manuscript page number (e.g. "Page X of 28")

Pp. 4-5: "Hills et al. (2010) argue that, with the structure of many search problems hinging on such persist-or-switch decisions, a domain-general mechanism mediating the explore-exploit tradeoff would be adaptive in humans" – this is not the only possibility – perhaps more likely is that a mechanism that originally evolved to solve the explore-exploit tradeoff in one domain (initially probably spatial) could be "exapted" in the course of evolution and used subsequently in a variety of other domains (e.g., in internal search).

P6: The Mekern (2017) study results *do* appear to fit with the priming idea, in that more exploitative local search in the spatial task – staying near where rewards have been found earlier,

in terms of greater turning angle after finding rewards – did correspond to longer search time in subsequent word puzzle patches. The difference from the Hills et al. 2008 study is that the Mekern spatial environments were constructed in a way that made the *diffuse* environments elicit more local search than the *clustered* environments. So the current authors' meta-analysis here should perhaps be redone by comparing results across the studies on the basis of the local versus global search *behavior* shown in the priming (spatial foraging) task, rather than the type of environment (patchy vs. diffuse) used in the priming task, which may not (as in Mekern's case) elicit the expected type of behavior.

In contrast to the Mekern study, the spatial environments in the current paper are constructed in a way that will probably maintain the same relationship between spatial structure (patchy vs. diffuse) and heightened local search (more in the patchy environments than in the diffuse ones) seen in the Hills et al. 2008 design (this is because the diffuse environments in the current design, similar to Hills et al., usually have only one rewarded location alone, rather than a small cluster). As a consequence, the analyses, which should be done in terms of both constructed spatial structure and observed amount of local search, will probably both point in the same direction with respect to the presence or absence of priming between the two search domains.

P10: Why was a measure of dispersion of searching across the *entire* search (all 50 tiles) used, rather than a measure of stepwise dispersion (from one selected tile to the next), as in the measures from Hills et al. and Mekern of turn angle from one step to the next? The global mean used here could obscure search patterns at the local level – for instance, the obtained value of $M=6.63$ in the clustered condition does not tell us anything directly about how far a participant moved from one tile to the next when searching (where presumably that distance is closer to 1-2). It is important to also measure the step-to-step clustering or dispersion (e.g. akin to turn angle or win-stay-lose-shift behavior) to assess whether the task and environment structure are successfully eliciting clustered versus diffuse search behavior in the participants, and hence are setting up the appropriate differences for the priming effect to be measured (as discussed above regarding the Mekern study where this appropriate elicitation did not happen). A related measure of local search behavior is used with a similar clicking grid search task in this paper:

Wilke, A., Minich, S., Panis, M., Langen, T.A., Skufca, J.D., & Todd, P.M. (2015). A game of hide and seek: Expectations of clumpy resources influence hiding and searching patterns. *PLoS ONE*, 10(7), e0130976. doi:10.1371/journal.pone.0130976.

P14: This seems like a reasonable design choice to not include a switching cost (time delay) between anagram "patches" (perhaps a downside could be some participants leaving all the patches too quickly and running out of patches, but that seems unlikely).

P17: Is there a reason for not having exactly 80 reward tiles per grid, so the overall available award is exactly equal across landscapes?

P18L52: "participants have 50 trials in which they can select any tile and gather points" –since the number of clicks in each landscape search is limited, rather than the time per search as in Hills et al. and Mekern, it would be good to also measure the time in seconds taken per landscape to see if that varies across environment types as a possible predictor of priming.

P18L42-45: "clustered landscapes consist of only one patch of resources so that participants are likely to continue exploiting. This will make the exploitation prime in our study more pronounced." –As discussed above, while having only one patch will increase the exploitation time compared to Hills et al.'s design, it will remove the need to monitor success and decide when to switch (leave a patch and explore for a new one). So different results in this study could possibly be attributed to lack of switching experience in the clustered landscapes.

P19L26-30: "The points that you can earn in each tile are from 0 to 100. Revealed tiles are also colour coded, as a visual aid to help you in this task. Darker colours correspond to larger rewards." –why talk about the points per tile as though there is a range of values, when there is actually only either 0 or 100 points per tile? It seems unnecessary to lead people to expect something that is not actually the case (i.e., that there are some intermediate reward values).

Decision letter (RSOS-201944.R0)

Dear Dr Anvari

On behalf of the Editors, I am pleased to inform you that your Manuscript RSOS-201944 entitled "Priming exploration across domains: Does search in a spatial environment influence search in a cognitive environment?" deemed suitable for in-principle acceptance in Royal Society Open Science subject to minor revision in accordance with the referee and editor suggestions. Please find their comments at the end of this email.

The reviewers and handling editors have recommended publication, but also suggest some minor revisions to your manuscript. Therefore, I invite you to respond to the comments and revise your manuscript.

Please you submit the revised version of your manuscript within 30 days (i.e. by the 5th February 2021). If you do not think you will be able to meet this date please let me know immediately.

Full author guidelines can be found here <https://royalsocietypublishing.org/rsos/registered-reports#ReviewerGuideRegRep>.

Kind regards
Professor Chris Chambers
Royal Society Open Science

on behalf of Professor Chris Chambers
(Subject Editor, Royal Society Open Science)
openscience@royalsociety.org

Associate Editor Comments to Author (Professor Chris Chambers):

Associate Editor: 1

Comments to the Author:

Two expert reviewers have now assessed the manuscript and provided very useful reviews. Both sets of comments are positive while also noting a number of detailed methodological and conceptual issues for consideration, including whether the study design and procedures are optimally calibrated to answering the research question. However, these concerns raised appear to be readily addressable through a careful revision and response.

Reviewer comments to Author:

Reviewer: 1

Comments to the Author(s)

The proposed research aims to replicate research of my own, with some improvements, focused around the question of whether or not there is a domain general cognitive search process that can be primed from one task to another. Naturally, having done the work myself once some time ago, I think the question is still interesting and the replication is important. The authors do an excellent job of describing what they plan to do and why, and I think barring a few minor corrections, it sounds great.

The Mekern et al study cited within as an example of a failure to replicate is really good work. I communicated with Vera Mekern and Bernhard Hommel about the work and some aspects of the design as well as the results. Several differences in the study are quite important, and I think, sadly, confuse the validity of the effect size comparison. First, the Eye-blink rate task is a 6-minute eye fixation task that takes place before and after the spatial foraging task. In Hills et al. 2010, we show that the priming effects we observed only lasts for about 3-minutes, mostly for the first two letter sets. So I wouldn't have expected the priming to survive the 6-minute EBR. Second, participants in Mekern turn more (with a higher turning angle) in the diffuse environment, which isn't what we observed in previous work and indeed would have likely been impossible to manage accurately in the 2008/10 work because the cursor was moving too quickly. I can't explain that difference, though Mekern suggests it may have had to do with the focused-meditation-like nature of the EBR. I really don't know. Possibly that led individuals to persevere on tiny patches in the diffuse task so they then primed themselves to persevere more in the scrabble task. Third, because of this flip in the spatial foraging task results, one could make flipped predictions about what should happen in the scrabble task as well, and of course that isn't helpful, not to mention the 6-minute EBR should have evened things out anyway. So that's all fiddly and complicated and I don't know what to make of it. One thing Vera's excellent work does show, however, is that the EBR is unaffected by the foraging perseveration in the spatial foraging task that precedes it. In this case, it doesn't matter how the spatial foraging task goes, as long as the conditions are different, then one would predict a difference in dopaminergic processes underlying search, if they indeed underly this search. The failure to find a difference in the EBR after the spatial foraging suggests this doesn't work via that mechanism. I am not an expert in EBR, but I trust that Hommel is, so I am inclined to believe that the mechanisms driving our results in 2008/10 may not work as previously proposed.

Back to the newly proposed research: There are a few ways in which the proposed paradigm diverges from the original. These are not necessarily criticisms and I think at least one is an improvement and necessary. However, a couple deserve a bit more thought.

1. Participants will be exposed to 14 letter sets before the spatial foraging task. In the original research (Hills, 2008,2010), this was substantially fewer in the pre-test. Arguably, 14 before and after is better for comparison, assuming that something unforeseen doesn't otherwise influence behavior (such as boredom or learning). There is no reason to expect these unforeseen things, so I'm inclined to think this is in an improvement.

2. Participants are able to submit three-letter words. The original uses four-letter words. I worry that three-letter words will allow participants to find solutions with less effort. This is a problem if one believes that the effortful part of the task is the part being primed, which I do. I vaguely recall spending some time on finding letter sets that provided enough four-letter solutions. Should this really matter? I'm not sure. In semantic memory search tasks, like category fluency, there is evidence of an executive search effect (e.g., Rosen & Engle, 1997). One would only expect to see this effect during effortful components of a search. If the search task is too easy, then it's not clear what to expect.

3. The proposed paradigm uses a 0-second time cost between letter sets. The original used a 15-second waiting period. The reason for this in the original research is because the task is designed to represent a patch-foraging task, similar to the spatial foraging task, and also similar to the marginal value theorem. According to the marginal value theorem, if there is no travel time between patches, and patch resource intake rates are always monotonically declining after arrival, then foragers should move between patches instantly. Obviously that's in the mathematical limit of no-travel costs whatsoever, which is unlikely to exactly match the anagrams. It takes time to read the letters and start thinking about how they could be manipulated. However, a 0-second waiting time invites participants to give up as soon as they encounter any difficulty, as the next patch will be a rich source of new solutions. Alongside the 3-letter words, I think this is likely to make the task too easy. Of course, it may not, but I feel it's better for me to point this out even if I'm wrong about how the task will work in practice. I guess I would ask how long it takes participants to find 30 items and how quickly do they switch between letter sets? If this comparable to the earlier work, then it is a better replication.

4. The grid search is the biggest difference from the prior work. I think this is an improvement. As the authors note, this tests not so much a replication as a test of the generalizability of the results to a very similar situation. It also potentially overcomes the arousal problem. Moreover, it avoids the problem observed in Mekern, that the spatial foraging manipulation didn't replicate.

5. In the analysis, will the analysis include a look within individual letter sets, as done in 2010?

6. Also, will the analysis include a focus on individual differences between the two tasks, following the last paragraph of the results in Hills et al., 2008? Neither 5. or 6. were done in Mekern's work, but I think they would have added value. Even if the manipulation failed to work as expected, the replication of a relationship between foraging externally and internally was still available for testing.

Overall, I think the design is a close replication with some minor variations and would be interesting and worthwhile. Plus the authors provide confidence that the work will be well done. The authors should consider whether they think the comments above warrant changing the design to limit solutions to only four or more letters and to increase the wait between new letter sets, as well as whether the final two comments (5 and 6) would further help address the underlying research question about a domain general search process.

Reviewer: 2

Comments to the Author(s)

Search for resources of different types is a crucial adaptive behavior for all mobile organisms, and there is ongoing debate about whether identical, similar, or different cognitive mechanisms are used for search in different domains. Priming studies between different search tasks have been used to inform this debate, but the results have been mixed, so a high-powered pre-registered replication would go far to advance this discussion and the understanding of search cognition.

This proposed study design looks to be a good way to replicate and test the generality of earlier search priming study results, including of Hills, Todd, and Goldstone (2008, 2010) and Mekern (2017). The authors make good arguments for the importance of replicating the earlier studies.

Their new design decisions are also appropriate, including increasing the power with a larger sample of participants, using a number of exclusion tests to ensure consistent sustained attention of the online participants, and having a fixed number of possible search actions per landscape to remove one possible alternative explanation of search priming results in terms of key-press based differences in arousal between the conditions. The methods are described in a clear and detailed manner that will allow others to readily replicate this study. The results of this study will be very useful in assessing the evidence for priming between different search domains, and consequently for pushing the field of search cognition forward.

There are though a few differences in this design from past study designs that should be considered if the results of this replication study are inconclusive. These are not arguments for changing the proposed design (aside from adding some additional measures and analyses, mentioned below in the detailed comments by page), but rather some further ideas about what could lead to priming between search tasks, if/when it does exist.

First, in the Hills et al. and Mekern spatial navigation search, and in the anagram task, it takes longer to make "bigger" steps (over the landscape, or to a very different word), but the current clicking search task is more like visual search, where longer steps across the landscape do not take (much) longer to produce (participants just have to move the mouse). Does that difference in the spatial task versions break down the connections between the two search tasks in some way?

Second, the fact that the current landscapes only have one big patch means there is no "exploit-to-explore" (patch-leaving) decisions to be made, which is different from the previous versions where there were multiple patches and so multiple switch-patch decisions. The anagram word puzzle includes multiple exploit-to-explore decisions, and in fact that is effectively the main thing being measured. How will the single patch landscapes change priming and behavior? (An alternative design would be to still have one patch, but of varying size across different landscapes, and let participants search as long as they want and switch to the next landscape whenever they want – this would invoke the exploit-to-explore decisions in the Hills et al. and Mekern designs, and in the anagram task.)

Third, because participants in the clustered landscape of the current design only explore one big patch continuously (once they find it) and get a near-constant stream of reward, they might therefore think the world is richer overall than do participants in the diffuse environment, because they have seen a much higher rate of return while searching. Would that different estimate of the "richness" of the world between the patchy and diffuse conditions affect behavior in the anagram task? If so, that could possibly mask any effect of the structure of the environment (clustered vs. diffuse). It could be that participants in the clustered environment

would be primed to think the world is very rich, e.g. full of big patches, which could lead them to then stay longer in each patch in the anagram task. Or possibly if they are primed to think the world is rich, then when they are in each anagram patch and they feel themselves slowing down in producing more words, they may think, “this *isn't* one of those big rich patches, so I should leave it right away and look for a big patch!” --leading to the opposite behavior, namely leaving sooner. Either way, if the single-cluster manipulation in the proposed study changes how rich the spatial world feels, that could make a difference in the results for an unintended reason.

Detailed comments by manuscript page number (e.g. “Page X of 28”)

Pp. 4-5: “Hills et al. (2010) argue that, with the structure of many search problems hinging on such persist-or-switch decisions, a domain-general mechanism mediating the explore-exploit tradeoff would be adaptive in humans” –this is not the only possibility – perhaps more likely is that a mechanism that originally evolved to solve the explore-exploit tradeoff in one domain (initially probably spatial) could be “exapted” in the course of evolution and used subsequently in a variety of other domains (e.g., in internal search).

P6: The Mekern (2017) study results *do* appear to fit with the priming idea, in that more exploitative local search in the spatial task – staying near where rewards have been found earlier, in terms of greater turning angle after finding rewards – did correspond to longer search time in subsequent word puzzle patches. The difference from the Hills et al. 2008 study is that the Mekern spatial environments were constructed in a way that made the *diffuse* environments elicit more local search than the *clustered* environments. So the current authors’ meta-analysis here should perhaps be redone by comparing results across the studies on the basis of the local versus global search *behavior* shown in the priming (spatial foraging) task, rather than the type of environment (patchy vs. diffuse) used in the priming task, which may not (as in Mekern’s case) elicit the expected type of behavior.

In contrast to the Mekern study, the spatial environments in the current paper are constructed in a way that will probably maintain the same relationship between spatial structure (patchy vs. diffuse) and heightened local search (more in the patchy environments than in the diffuse ones) seen in the Hills et al. 2008 design (this is because the diffuse environments in the current design, similar to Hills et al., usually have only one rewarded location alone, rather than a small cluster). As a consequence, the analyses, which should be done in terms of both constructed spatial structure and observed amount of local search, will probably both point in the same direction with respect to the presence or absence of priming between the two search domains.

P10: Why was a measure of dispersion of searching across the *entire* search (all 50 tiles) used, rather than a measure of stepwise dispersion (from one selected tile to the next), as in the measures from Hills et al. and Mekern of turn angle from one step to the next? The global mean used here could obscure search patterns at the local level – for instance, the obtained value of $M=6.63$ in the clustered condition does not tell us anything directly about how far a participant moved from one tile to the next when searching (where presumably that distance is closer to 1-2). It is important to also measure the step-to-step clustering or dispersion (e.g. akin to turn angle or win-stay-lose-shift behavior) to assess whether the task and environment structure are successfully eliciting clustered versus diffuse search behavior in the participants, and hence are setting up the appropriate differences for the priming effect to be measured (as discussed above regarding the Mekern study where this appropriate elicitation did not happen). A related measure of local search behavior is used with a similar clicking grid search task in this paper:

Wilke, A., Minich, S., Panis, M., Langen, T.A., Skufca, J.D., & Todd, P.M. (2015). A game of hide and seek: Expectations of clumpy resources influence hiding and searching patterns. *PLoS ONE*, 10(7), e0130976. doi:10.1371/journal.pone.0130976.

P14: This seems like a reasonable design choice to not include a switching cost (time delay) between anagram “patches” (perhaps a downside could be some participants leaving all the patches too quickly and running out of patches, but that seems unlikely).

P17: Is there a reason for not having exactly 80 reward tiles per grid, so the overall available award is exactly equal across landscapes?

P18L52: “participants have 50 trials in which they can select any tile and gather points” –since the number of clicks in each landscape search is limited, rather than the time per search as in Hills et al. and Mekern, it would be good to also measure the time in seconds taken per landscape to see if that varies across environment types as a possible predictor of priming.

P18L42-45: “clustered landscapes consist of only one patch of resources so that participants are likely to continue exploiting. This will make the exploitation prime in our study more pronounced.” –As discussed above, while having only one patch will increase the exploitation time compared to Hills et al.’s design, it will remove the need to monitor success and decide when to switch (leave a patch and explore for a new one). So different results in this study could possibly be attributed to lack of switching experience in the clustered landscapes.

P19L26-30: “The points that you can earn in each tile are from 0 to 100. Revealed tiles are also colour coded, as a visual aid to help you in this task. Darker colours correspond to larger rewards.” –why talk about the points per tile as though there is a range of values, when there is actually only either 0 or 100 points per tile? It seems unnecessary to lead people to expect something that is not actually the case (i.e., that there are some intermediate reward values).

Author's Response to Decision Letter for (RSOS-201944.R0)

See Appendix A.

RSOS-201944.R1 (Revision)

Review form: Reviewer 1

Do you have any ethical concerns with this paper?

No

Recommendation?

Accept in principle

Comments to the Author(s)

Overall, I appreciate the authors attention to detail and response to the previous comments. I think the proposed study will test a general form of the hypothesis that 'search in one domain influences search in another domain', which the authors claim is their primary goal.

They are also testing hypotheses about how that search process might work. For example, one question is whether or not the mechanism is meta-cognitive: people realize they are searching

more locally or more globally and then unwittingly apply this 'frame' in a new domain. Or whether it is more procedural: Some low level 'turning/look-back' process is activated and applied across domains. Moreover, the adjustment in turning rate may be a function of how far the individual has to travel between patches or how long it takes. If it's more meta-cognitive, then knowing that letter sets have no travel time between them may be important, as this may be integrated into the overall strategy. If the authors have found that people still spend sufficient time in letter sets with no travel time between them, then this shouldn't be a problem. If it's a low-level turning process that is calibrated (e.g., like it might be for the Margial Value Theorem, see Adler and Kotar), then the time to move from one place to another in the spatial landscape may be important. It will be interesting to know that and this can possibly be tested in the exploratory analysis as well.

I agree with the authors that a very general form of the hypothesis is not based on these subtleties about the underlying process.

Minor comment: I wouldn't say 'duplicated' as on p. 16 Line 13. This has a very specific biological/genetic meaning, which often means it can act independently from the copy on which it is based. We're talking about behaviour here, so the semantic mapping is perhaps a bit looser, but 'exapted' is probably the word the authors want and the one truer to the original Hills et al hypothesis. Exapted simply means the process was adapted to a new purpose. In fact, I've often felt it is the same purpose merely confused by the fact that people tend to think there's a difference between external information and internal information. There is a difference, but it may not be one that is meaningful to our search processes, however they may work.

Review form: Reviewer 2

Do you have any ethical concerns with this paper?

No

Recommendation?

Accept in principle

Comments to the Author(s)

The authors have done a thorough job of responding to the two reviewers' comments and have argued well for their experimental design. I have only a few remaining comments where the authors can consider further explanatory possibilities and exploratory measures and analyses, in response to the indicated locations in the revised manuscript. I look forward to seeing the results of this replication and continuing the discussion.

P7L22-27: "Because in our design the clustered condition involves grids with one large patch, participants should be more strongly primed to continue exploiting the one patch, or, put another way, to explore in a more clustered way, than if we had grids with multiple patches." - I disagree with the authors' argument here. Search in a clustered environment can lead to later behavioral differences (i.e., can have a priming effect) compared to search in a dispersed environment only if those two searches are different; and what can make those two searches different is having to deal with clusters in the former and not in the latter – for instance by finding clusters, staying in them, and leaving them when it's appropriate to do so. In the current design with one big cluster, there is little finding to be done, relatively little need to do anything to stay inside the big cluster, and no leaving decisions to be made – making the appropriate behaviors here not so different from those for the dispersed environment. (If the single cluster size were increased, these differences would decline toward zero.) So this does not seem to me to be a design where

priming will be stronger, but rather weaker. The question is whether the low levels of these differences will be sufficient to induce observable levels of priming (which will be interesting), but if no priming is found, it may be a consequence of not using environments of the sort where such priming should be expected.

P9: With regard to Mekern's results, the hypothesis is *not* that "search in a spatial environment will prime search in a cognitive environment" - it's that the *type* of search that an organism does in a spatial environment will prime the same type of search in a cognitive environment, and that is also what Mekern's results show, when analyzed in terms of the type of search, not the type of environment. So the suggestion is to do the analysis of this new proposed study, and the meta-analysis of all the studies being compared, on that basis, of search type, rather than of environment type.

P10L6ff: On the dispersion measures to use, I appreciate the authors' argument that the local dispersion measure could count some instances where the searcher jumped to another part of the same patch as being non-local - but the usefulness of that measure would be to assess local versus non-local movement, not necessarily within-patch versus not-within-patch movement, and this would still be useful to know for the reasons spelled out in my previous review regarding distinguishing different types of search behavior. Hence reporting both measures (even if one is exploratory) would help researchers to understand better what people are doing in the different search environments.

P11L41ff: It will be interesting to look (exploratorily) at the time taken per grid to see if that predicts the amount of priming found at the individual level in the different environment types, and also whether it is correlated with the dispersion for each grid - does it take more time (and maybe more thought) to search dispersed environments than clustered ones? Do people who search clustered environments in a more dispersed manner take longer and show less priming of search in the anagram task?

Decision letter (RSOS-201944.R1)

Dear Dr Anvari,

On behalf of the Editors, I am pleased to inform you that your Manuscript RSOS-201944.R1 entitled "Priming exploration across domains: Does search in a spatial environment influence search in a cognitive environment?" has been accepted in principle for publication in Royal Society Open Science subject to minor revision in accordance with the referee and editor suggestions. Please find their comments at the end of this email.

The reviewers and handling editors have recommended publication, but also suggest some minor revisions to your manuscript. Therefore, I invite you to respond to the comments and revise your manuscript.

Please you submit the revised version of your manuscript within 7 days (i.e. by the 25-Feb-2021). If you do not think you will be able to meet this date please let me know immediately.

When submitting your revised manuscript, you will be able to respond to the comments made by the referees and you should upload a file "Response to Referees". You can use this to document any changes you make to the original manuscript. In order to expedite the processing of the revised manuscript, please be as specific as possible in your response to the referees.

Full author guidelines can be found here <https://royalsocietypublishing.org/rsos/registered-reports>.

on behalf of Professor Chris Chambers (Subject Editor, Royal Society Open Science)
openscience@royalsociety.org

Associate Editor Comments to Author (Professor Chris Chambers):

Both reviewers from the previous round returned to assess the revised Stage 1 manuscript. Both now recommend IPA following some remaining minor revisions to address conceptual concerns and potential design limitations. Any points regarding exploratory unregistered analyses can be addressed at Stage 2. If the authors are able to address these remaining issues in a final Stage 1 revision then IPA will be forthcoming without requiring further in-depth review.

Reviewer comments to Author:

Reviewer: 1

Comments to the Author(s)

Overall, I appreciate the authors attention to detail and response to the previous comments. I think the proposed study will test a general form of the hypothesis that 'search in one domain influences search in another domain', which the authors claim is their primary goal.

They are also testing hypotheses about how that search process might work. For example, one question is whether or not the mechanism is meta-cognitive: people realize they are searching more locally or more globally and then unwittingly apply this 'frame' in a new domain. Or whether it is more procedural: Some low level 'turning/look-back' process is activated and applied across domains. Moreover, the adjustment in turning rate may be a function of how far the individual has to travel between patches or how long it takes. If it's more meta-cognitive, then knowing that letter sets have no travel time between them may be important, as this may be integrated into the overall strategy. If the authors have found that people still spend sufficient time in letter sets with no travel time between them, then this shouldn't be a problem. If it's a low-level turning process that is calibrated (e.g., like it might be for the Margial Value Theorem, see

Adler and Kotar), then the time to move from one place to another in the spatial landscape may be important. It will be interesting to know that and this can possibly be tested in the exploratory analysis as well.

I agree with the authors that a very general form of the hypothesis is not based on these subtleties about the underlying process.

Minor comment: I wouldn't say 'duplicated' as on p. 16 Line 13. This has a very specific biological/genetic meaning, which often means it can act independently from the copy on which it is based. We're talking about behaviour here, so the semantic mapping is perhaps a bit looser, but 'exapted' is probably the word the authors want and the one truer to the original Hills et al hypothesis. Exapted simply means the process was adapted to a new purpose. In fact, I've often felt it is the same purpose merely confused by the fact that people tend to think there's a difference between external information and internal information. There is a difference, but it may not be one that is meaningful to our search processes, however they may work.

Reviewer: 2

Comments to the Author(s)

The authors have done a thorough job of responding to the two reviewers' comments and have argued well for their experimental design. I have only a few remaining comments where the authors can consider further explanatory possibilities and exploratory measures and analyses, in response to the indicated locations in the revised manuscript. I look forward to seeing the results of this replication and continuing the discussion.

P7L22-27: "Because in our design the clustered condition involves grids with one large patch, participants should be more strongly primed to continue exploiting the one patch, or, put another way, to explore in a more clustered way, than if we had grids with multiple patches." - I disagree with the authors' argument here. Search in a clustered environment can lead to later behavioral differences (i.e., can have a priming effect) compared to search in a dispersed environment only if those two searches are different; and what can make those two searches different is having to deal with clusters in the former and not in the latter – for instance by finding clusters, staying in them, and leaving them when it's appropriate to do so. In the current design with one big cluster, there is little finding to be done, relatively little need to do anything to stay inside the big cluster, and no leaving decisions to be made – making the appropriate behaviors here not so different from those for the dispersed environment. (If the single cluster size were increased, these differences would decline toward zero.) So this does not seem to me to be a design where priming will be stronger, but rather weaker. The question is whether the low levels of these differences will be sufficient to induce observable levels of priming (which will be interesting), but if no priming is found, it may be a consequence of not using environments of the sort where such priming should be expected.

P9: With regard to Mekern's results, the hypothesis is *not* that "search in a spatial environment will prime search in a cognitive environment" - it's that the *type* of search that an organism does in a spatial environment will prime the same type of search in a cognitive environment, and that is also what Mekern's results show, when analyzed in terms of the type of search, not the type of environment. So the suggestion is to do the analysis of this new proposed study, and the meta-analysis of all the studies being compared, on that basis, of search type, rather than of environment type.

P10L6ff: On the dispersion measures to use, I appreciate the authors' argument that the local dispersion measure could count some instances where the searcher jumped to another part of the same patch as being non-local - but the usefulness of that measure would be to assess local versus non-local movement, not necessarily within-patch versus not-within-patch movement,

and this would still be useful to know for the reasons spelled out in my previous review regarding distinguishing different types of search behavior. Hence reporting both measures (even if one is exploratory) would help researchers to understand better what people are doing in the different search environments.

P11L41ff: It will be interesting to look (exploratorily) at the time taken per grid to see if that predicts the amount of priming found at the individual level in the different environment types, and also whether it is correlated with the dispersion for each grid – does it take more time (and maybe more thought) to search dispersed environments than clustered ones? Do people who search clustered environments in a more dispersed manner take longer and show less priming of search in the anagram task?

Author's Response to Decision Letter for (RSOS-201944.R1)

See Appendix B.

Decision letter (RSOS-201944.R2)

Dear Dr Anvari

On behalf of the Editor, I am pleased to inform you that your Manuscript RSOS-201944.R2 entitled "Priming exploration across domains: Does search in a spatial environment influence search in a cognitive environment?" has been accepted in principle for publication in Royal Society Open Science.

You may now progress to Stage 2 and complete the study as approved. Before commencing data collection we ask that you:

- 1) Update the journal office as to the anticipated completion date of your study.
- 2) Register your approved protocol on the Open Science Framework (<https://osf.io/>) or other recognised repository, either publicly or privately under embargo until submission of the Stage 2 manuscript. Please note that a time-stamped, independent registration of the protocol is mandatory under journal policy, and manuscripts that do not conform to this requirement cannot be considered at Stage 2. The protocol should be registered unchanged from its current approved state, with the time-stamp preceding implementation of the approved study design. We recommend using the dedicated Stage 1 RR registration portal provided by the OSF at <https://osf.io/rr>

Following completion of your study, we invite you to resubmit your paper for peer review as a Stage 2 Registered Report. Please note that your manuscript can still be rejected for publication at Stage 2 if the Editors consider any of the following conditions to be met:

- The results were unable to test the authors' proposed hypotheses by failing to meet the approved outcome-neutral criteria.
- The authors altered the Introduction, rationale, or hypotheses, as approved in the Stage 1 submission.
- The authors failed to adhere closely to the registered experimental procedures. Please note that any deviations from the approved experimental procedures must be communicated to the editor immediately for approval, and prior to the completion of data collection. Failure to do so can result in revocation of in-principle acceptance and rejection at Stage 2 (see complete guidelines for further information).
- Any post-hoc (unregistered) analyses were either unjustified, insufficiently caveated, or overly dominant in shaping the authors' conclusions.
- The authors' conclusions were not justified given the data obtained.

We encourage you to read the complete guidelines for authors concerning Stage 2 submissions at <https://royalsocietypublishing.org/rsos/registered-reports#ReviewerGuideRegRep>. Please especially note the requirements for data sharing, reporting the URL of the independently registered protocol, and that withdrawing your manuscript will result in publication of a Withdrawn Registration.

Once again, thank you for submitting your manuscript to Royal Society Open Science and we look forward to receiving your Stage 2 submission. If you have any questions at all, please do not hesitate to get in touch. We look forward to hearing from you shortly with the anticipated submission date for your stage two manuscript.

on behalf of Professor Chris Chambers (Registered Reports Editor, Royal Society Open Science)
openscience@royalsociety.org

Author's Response to Decision Letter for (RSOS-201944.R2)

See Appendix C.

RSOS-201944.R3 (Revision)

Review form: Reviewer 1

Is the manuscript scientifically sound in its present form?

Yes

Are the interpretations and conclusions justified by the results?

Yes

Is the language acceptable?

Yes

Do you have any ethical concerns with this paper?

No

Have you any concerns about statistical analyses in this paper?

No

Recommendation?

Accept with minor revision

Comments to the Author(s)

This is an impressive piece of work. The authors show in a convincing way that priming of exploration and exploitation across domains does not extend to their paradigm, which presents some useful questions for further study in this area. Specifically, by attempting to prime exploration/exploitation from a spatial to a more internal cognitive domain (the anagram task), the authors demonstrate in a large pre-registered experiment, with some modifications from the original experiment, the absence of statistical evidence for a priming effect above a small effect size. The modifications are purposeful and invite questions about the boundary conditions for such priming.

Overall, the work is well-written, it is well-designed, the analyses are clear and straightforward, and the results are compelling. As the first author of the original work on this topic, I welcome this research and I think the authors do a good job of helping highlight the questions one would need to address to fully understand cross-domain priming. I had not fully appreciated the numerous decisions involved in evaluating this hypothesis and working through this article has helped to clarify my thinking--some of which I've written below. The authors have done an excellent job here and I think their analyses and discussion are well motivated and fair.

I have a few minor comments below. I say they are minor because I don't think any of them dispute the claims and interpretation made in the article. A couple at the beginning are minor corrections about prior theorizing. Otherwise, these are mostly for my own curiosity and, though probably most can be addressed with the open source data provided, I suspect some of them may help guide future research even for those who don't go delving into the data. Nonetheless, I leave it to the editor and authors to decide whether to include these in the work or not.

1. The authors say a few times that the previous work posits "a domain-general tendency to explore". I think the correct claim--the one most strongly supported by the introduction in Hills et al., 2010--is that there is a "domain general search process". That hypothesized process is likely to operate across domains and therefore be susceptible to priming. Specifically, on p. 4, line 26, and p.31 line 7, I think the wording should change to reflect that what is being primed is a domain general search process. The claim for a tendency to explore is less clear for several reasons. First, explore must be juxtaposed against the alternative: exploit, which would suggest there is not a tendency to exploit. That seems untrue in general, but it is also not the claim of the prior work.

2. The theory behind the domain general search process is largely based on the role of executive function across domains and its putative evolution from a prior spatial search process. The notion of cross-domain priming hinges on the potential that similar processes underlying executive function are also used across domains. We do know that measures of executive function correlate with behavior across many domains, many of which (possibly all) can be characterized as search processes, or evolutionary descendants of explore/exploit processes (e.g., goal maintenance,

giving up times, etc.). The question is whether the search processes can be primed across domains. In light of the present results, I see three possibilities: 1) executive function is not a single entity that is domain general, but rather a host of subprocesses that are recruited for different tasks. There is therefore nothing to prime. This is consistent with a more modular understanding of executive function--with no common elements associated with search across tasks. 2) Search processes underlying executive function are shared across domains but they aren't primable. Or 3) Search processes are shared across tasks but they rapidly adapt to new task demands and represent a small influence that is easily modified by other task demands. Indeed, on this final point, they may adapt or respond especially rapidly to situations where the 'right' level of mediating between exploration and exploitation is already known. I think the authors results are consistent with any of these three. If we incorporate the previous evidence for priming, this would favor 3). But I would not put 1) or 2) out of the question. As the authors note, there are other reasons why we may have seen priming in the previous research.

3. A couple of times the article mentions an adaptational argument behind the basis for the priming hypothesis. This isn't clear. E.g., the work by Hills et al 2010 argues that the evolutionary basis for the shared process is likely to be one of exaptation, borrowing an existing spatial search process to search in an interior domain. I think the authors make a novel and more interesting claim but attribute it to Hills et al 2010 when they write: (p4 l 60) "Hills et al. (2010) argue that, with the structure of many search problems hinging on such persist-or-switch decisions, a domain-general mechanism mediating the explore-exploit tradeoff would be adaptive in humans." To be clear, Hills et al do argue that the existence of a search process, that can modulate between exploration and exploitation, is adaptive. But the domain generality is not argued to be adaptive, rather it is informed speculation based on the evidence for shared neuromolecular architecture in comparative biological research across domains and the evidence as explained above about executive function. I wouldn't claim these are adaptive. Indeed, I think a strongly primable search process would be maladaptive, unless we had some reason to believe that tuning the search process for one domain would inform the tuning for a subsequent very different domain. That seems unlikely. Search should be tuned to the environment in which it finds itself. Nonetheless, there may be an adaptive argument here similar to the one proposed in the article and I'm just missing it. If this can be made clearer that would be helpful.

4. I would be curious to hear more about the 50 second warning. I am curious if this 'primes' participants to signal their activity during the anagram task, which they can only do by entering words or moving forward in the task. The instructions say "if you are inactive for a substantial period of time", but I don't think the participants are told for how long, and it also says you'll be financially penalized if you are inactive for longer than the undisclosed amount of time. I'm not sure whether this would cause people to move on from one letter set to another--I suspect it would for some. Obviously, everyone is affected by this equally, but it may nonetheless be a stronger motivation to move on than any cross-domain priming effect. This isn't a criticism: for an online task (or even for a laboratory task) one needs to ensure the participants are fresh and focused. This may have been achieved in the prior studies by only having a short set of pre-test anagrams. Here it is achieved by incentivizing activity.

5. Do people get better at the anagram task as they go on? That is, do they find more words at a faster rate or more words per letter set? Does this settle down before the post-test, suggesting they have an established expectation/strategy? Initially I thought this couldn't be the case, because the difference is negative for both conditions in Figure 2. But in fact it may be that the first four or five letter sets in the pre-test are enough to find a stable strategy (e.g., "find n words and move on"). Those using such a strategy might show a negative patch time between pre and post, assuming that the first few letter sets took longer, in order to find a reasonable point for switching. I guess I am asking why they are taking less time per letter set after the task. Are they getting better at solving anagrams, calibrating the time per letter set, or simply moving on more

quickly, e.g., because they received a warning? Do people who receive a warning change their behavior by moving on more quickly?

6. Because there is no travel time between patches, people will be more inclined to move to the next patch than if the travel time were longer. This is the basic prediction, over many studies, of the marginal value theorem. This isn't necessarily a problem in the present study, assuming that it takes time to warm up to a patch and this time is longer than the time it takes to find the second item after finding the first. Otherwise, the optimal strategy is always to move to the next patch immediately after finding one item. Of course, people may not realize this or act 'optimally' based on some other criterion than rate. I don't think this is a problem here, as the authors exclude people who found less than 20 words, but it might mean that people were strongly incentivized to move on from one patch to the next (in addition to the potential warning for non-activity mentioned above). The authors might want to report the average time to find the first item. More interesting would be to report the average time to find all items (first, second, third, and so on), average number of items found per patch, and how this changed across patches.

7. On p. 24, l 17, the authors write that "only those who find 20 words in both pre-test and the post-test sessions will be included in the analysis." Does this mean that not all participants found 30 words in the pretest or the posttest? I may have missed it, but I guess this is because if they get through all 14 patches, then they can also complete the anagram task – or is there another reason?

8. 150 seconds on the post-test anagram instructions seems quite long. Did many participants take longer than 20 or so seconds to read these instructions? They had already seen them once before, right? I'm guessing this is a long-tailed distribution. In a laboratory this might not be much of a problem (they might not have phones or other distractions such as email to briefly check), but I am naturally curious if this matters more for an online experiment. Nor am I sure what the right thing to do here is.

In sum, I think this is good and thorough work. Methodologically, it exemplifies good research and a state-of-the-art open science approach. Scientifically, the work raises interesting questions about the role and primability of a common search process for exploration and exploitation across domains.

Review form: Reviewer 2

Is the manuscript scientifically sound in its present form?

Yes

Are the interpretations and conclusions justified by the results?

Yes

Is the language acceptable?

Yes

Do you have any ethical concerns with this paper?

No

Have you any concerns about statistical analyses in this paper?

No

Recommendation?

Accept with minor revision

Comments to the Author(s)

Review of “Priming exploration across domains: Does search in a spatial environment influence search in a cognitive environment?” registered report stage 2 (RSOS-201944.R3), submitted to Royal Society Open Science

Reviewed by Peter M. Todd, Indiana University Cognitive Science Program (original Reviewer 2)

This full version of the paper, reporting the pre-registered and non-pre-registered analyses on the exploration priming study results, is thorough and well-done. It is useful to see the more detailed meta-analyses with Mekern et al.’s data interpreted both in favor of and against the priming hypothesis. The modification of the spatial environments to include multiple patches is helpful for making this study design more readily comparable to that of Hills et al. 2008/2010. The additional non-pre-registered analyses, enabling consideration of the effect of search behavior (in both the meta-analysis and the current experiment) in addition to the effect of environment structure, are important for trying to understand the differing results between this study and previous research. The authors lay out a range of possible explanations and appropriate strategies for exploring them further. All of this is very beneficial for helping us gain a clearer understanding of search cognition.

One additional measure that could be informative to add to the current non-pre-registered analyses is to see whether environment structure condition differentially affects how people respond after finding or not finding a resource in the spatial search task. Searchers in clumpy environments should stay “local” after finding a resource to increase their chances of finding other resources expected nearby, while searchers in dispersed environments should move away after finding a resource, searching more globally rather than staying local where other resources would not be expected. This was found in Hills et al. 2008 (Figure 2) in terms of turning angle, and in Wilke et al. (2015) in terms of distance jumped in a grid-search task (where win-stay versus win-shift behavior was seen). It could be that the proposed priming effect depends on resource-dependent shifting between local and global search that is common in foraging; if so, then testing for the presence of such shifting in the current data would help further constrain when priming could occur (the test would involve dividing the stepwise clustering measure used on manuscript page 26 into two measures, for those steps following finding a resource versus not finding a resource, and comparing those two measures across the two conditions).

Reference:

Wilke, A., Minich, S., Panis, M., Langen, T.A., Skufca, J.D., & Todd, P.M. (2015). A game of hide and seek: Expectations of clumpy resources influence hiding and searching patterns. *PLoS ONE*, 10(7), e0130976. doi:10.1371/journal.pone.0130976.

An experiment design difference that should be considered in further follow-up studies is the ability to “jump” equally easily anywhere in this grid search task, versus having to move continuously between locations in Hills et al.’s spatial search design. In the current study, local and global search “cost” roughly the same in terms of effort and time, while in Hills et al., global search takes longer to achieve, having to keep moving longer to get further across the grid (though local search takes more actions, in terms of key-presses, to turn and stay in a local area rather than moving more “straight ahead”, away from the previous location, for global search). This difference between the two spatial task versions in terms of distance and travel time may matter for possible priming.

One other difference that may be worth considering further, this time in the results rather than design, is that in Hills et al. 2008 the clumpy spatial environments *increased* the time spent in letter clusters while dispersed spatial environments *decreased* the time spent in letter clusters

(Figure 3), but in the current study *both* environment types *decreased* the time spent in letter clusters. Why might that be?

Overall, the data reported are able to test the authors' hypothesis, which is the same as in the Stage 1 submission; the authors followed the registered experimental procedures aside from a couple of small changes that they received approval for and note in the paper; the exploratory analyses are justified, sound, and informative; and the conclusions are justified given the data.

Detailed comments by page:

Throughout: contractions (e.g. doesn't, didn't, we're, they'd) should be spelled out.

Manuscript page 3 (in lower right-hand corner), lines 26-28: "there may be a domain-general tendency to explore, possibly due to common search processes being involved in different domains (e.g., Hills & Ducas, 2012; Wilke et al., 2009)." -the phrase "domain-general tendency to explore" here is ambiguous and should be clarified. It could mean that most organisms tend to explore (as opposed to exploit) in most domains they encounter (i.e., exploration is common, across species and across domains). Or it could mean that any given organism will have some individually-specific tendency to explore, from low to high, and that tendency will be similarly manifest (e.g. usually low or usually high) across the various domains that this organism explores in (i.e., individual differences in exploration tendency across domains, as assessed – but not found – by von Helversen et al. 2018 and as explored by the authors in this paper on P27L25-51). Or it could mean that search itself (including the exploration/exploitation tradeoff) is done in similar ways across species and across domains. The last interpretation seems most likely here, but given the ambiguity, this sentence should probably be reworded to be more specific, e.g. "there may be a domain-general approach to exploration and exploitation across species, possibly due to..." A similar rewording would apply to the same phrase used on P30L7-11 (in two locations).

Reference:

von Helversen, Bettina; Mata, Rui; Samanez-Larkin, Gregory R; Wilke, Andreas (2018). Foraging, exploration, or search? On the (lack of) convergent validity between three behavioral paradigms. *Evolutionary Behavioral Sciences*, 12(3):152-162. DOI: <https://doi.org/10.1037/ebs0000121>

P3L28 and P30L4: "Ducas" should be "Dukas"

P8L39: talking about letter-set timing here, before the cognitive task has been defined, may be confusing – can this be phrased more abstractly, e.g. in terms of patches in the cognitive task? Or describe the tasks briefly before indicating the hypotheses?

P9L53: "averaging this across all 5 grids" - the task has not been described yet, so to make this clearer add e.g. "that each participant searched"

P11L35-47: The participant numbers do not seem to add up properly: $400 - 47 - 2 = 351$, but 347 are indicated; and then there are 174 in the clustered condition and 171 in the dispersed condition, which is 345. And just to confirm, there were nearly twice as many female (221) as male participants (~120)?

P14: Were the participants shown how many words they had generated overall, so they knew how much further they had to go (as was shown for the grid search)? Did they see the words they had already generated from the current letter-set displayed on the screen? A screen-shot of this task as well could be useful.

P23L17: "(i.e., only those who find 20 words" should be "...who find at least 20 words"

P23L56-58: “But to complete the task participants needed to find 30 words and to be included in the analyses they needed to find 20 words.” – this is confusing – what happens if a participant does not complete the task? Do they not get paid? Do they not get to proceed to the next part of the study? Or are there no consequences? And, whatever these consequences of not completing are, their data is still analyzed if they found 20 or more words? And how is “not completing” defined – does it mean that they reach letter-cluster 14 and ask for the next cluster (which presumably does not exist) without having found 30 words?

How many participants did the above classifications apply to? And how many received delay warnings or were removed because of delays or failing comprehension questions?

P27L9: “People in the clustered condition spent fewer minutes in each landscape ($M = 5.56$,” – “each landscape” should be “these landscapes”, right? That is, people presumably did not spend 5 minutes in each landscape, or else they would be spending 25 minutes on the grid search task overall (and the whole study took ~28 minutes, as indicated on P11).

In the data file, the two environment types are called “rough” and “smooth” – in the Priming Search Codebook file it would be useful to indicate the mapping of these types onto clustered and dispersed.

Decision letter (RSOS-201944.R3)

Dear Dr Anvari:

On behalf of the Editor, I am pleased to inform you that your Stage 2 Registered Report RSOS-201944.R3 entitled "Priming exploration across domains: Does search in a spatial environment influence search in a cognitive environment?" has been deemed suitable for publication in Royal Society Open Science subject to minor revision in accordance with the referee suggestions. Please find the referees' comments at the end of this email.

The reviewers and Subject Editor have recommended publication, but also suggest some minor revisions to your manuscript. Therefore, I invite you to respond to the comments and revise your manuscript.

Please also ensure that all the below editorial sections are included where appropriate -- if any section is not applicable to your manuscript, please can we ask you to nevertheless include the heading, but explicitly state that the heading is inapplicable. An example of these sections is attached with this email.

- Ethics statement

- Data accessibility

[http://datadryad.org/submit?journalID=RSOS&manu=\(Document not available\)](http://datadryad.org/submit?journalID=RSOS&manu=(Document not available))

- Competing interests

- Authors' contributions

- Acknowledgements

- Funding statement

Because the schedule for publication is very tight, it is a condition of publication that you submit the revised version of your manuscript within 21 days (i.e. by the 16-Jul-2021). If you do not think you will be able to meet this date please let me know immediately.

on behalf of Professor Chris Chambers
(Registered Reports Editor, Royal Society Open Science)
openscience@royalsociety.org

Associate Editor Comments to Author (Professor Chris Chambers):

Associate Editor: 1

Comments to the Author:

The Stage 2 manuscript was evaluated by the two reviewers who assessed it at Stage 1. As you will see, both reviews are very positive while also raising a number of very thoughtful and constructive points to consider, including clarifications, additional discussion points and potential additional analyses.

I want to offer some editorial guidance here to ensure that the revision meets both the spirit and letter of the RR policy. First, please keep to a minimum any changes to the approved Stage 1 part of the manuscript. Obviously, any factual errors should be corrected, but elaboration or reframing of points of interpretation in the Introduction should be reserved for the Discussion, or at most, if vital for avoiding confusion, footnoted in the Introduction as a point raised during Stage 2 review. Second, the ideas from the reviewers concerning additional exploratory analyses strike me as interesting and well worth considering; however the inclusion of additional analyses is not required for Stage 2 RRs unless the analyses are judged necessary to justify the current conclusions. In this case, I believe that test is not met (even if the additional analyses are of interest); therefore, while I would encourage you to consider the merit of these suggestions very carefully, it is your choice as authors whether to report these additional analyses.

Provided you are able to respond thoroughly to the reviewers' comments, Stage 2 acceptance should be forthcoming without requiring further in-depth review.

Comments to Author:

Reviewer: 1

Comments to the Author(s)

This is an impressive piece of work. The authors show in a convincing way that priming of exploration and exploitation across domains does not extend to their paradigm, which presents some useful questions for further study in this area. Specifically, by attempting to prime exploration/exploitation from a spatial to a more internal cognitive domain (the anagram task), the authors demonstrate in a large pre-registered experiment, with some modifications from the original experiment, the absence of statistical evidence for a priming effect above a small effect size. The modifications are purposeful and invite questions about the boundary conditions for such priming.

Overall, the work is well-written, it is well-designed, the analyses are clear and straightforward, and the results are compelling. As the first author of the original work on this topic, I welcome this research and I think the authors do a good job of helping highlight the questions one would need to address to fully understand cross-domain priming. I had not fully appreciated the numerous decisions involved in evaluating this hypothesis and working through this article has helped to clarify my thinking--some of which I've written below. The authors have done an excellent job here and I think their analyses and discussion are well motivated and fair.

I have a few minor comments below. I say they are minor because I don't think any of them dispute the claims and interpretation made in the article. A couple at the beginning are minor corrections about prior theorizing. Otherwise, these are mostly for my own curiosity and, though probably most can be addressed with the open source data provided, I suspect some of them may help guide future research even for those who don't go delving into the data. Nonetheless, I leave it to the editor and authors to decide whether to include these in the work or not.

1. The authors say a few times that the previous work posits "a domain-general tendency to explore". I think the correct claim--the one most strongly supported by the introduction in Hills et al., 2010--is that there is a "domain general search process". That hypothesized process is likely to operate across domains and therefore be susceptible to priming. Specifically, on p. 4, line 26, and p.31 line 7, I think the wording should change to reflect that what is being primed is a domain general search process. The claim for a tendency to explore is less clear for several reasons. First, explore must be juxtaposed against the alternative: exploit, which would suggest there is not a tendency to exploit. That seems untrue in general, but it is also not the claim of the prior work.

2. The theory behind the domain general search process is largely based on the role of executive function across domains and its putative evolution from a prior spatial search process. The notion of cross-domain priming hinges on the potential that similar processes underlying executive function are also used across domains. We do know that measures of executive function correlate with behavior across many domains, many of which (possibly all) can be characterized as search processes, or evolutionary descendants of explore/exploit processes (e.g., goal maintenance, giving up times, etc.). The question is whether the search processes can be primed across domains. In light of the present results, I see three possibilities: 1) executive function is not a single entity that is domain general, but rather a host of subprocesses that are recruited for different tasks. There is therefore nothing to prime. This is consistent with a more modular understanding of executive function--with no common elements associated with search across tasks. 2) Search processes underlying executive function are shared across domains but they aren't primable. Or 3) Search processes are shared across tasks but they rapidly adapt to new task demands and represent a small influence that is easily modified by other task demands. Indeed, on this final point, they may adapt or respond especially rapidly to situations where the 'right' level of mediating between exploration and exploitation is already known. I think the authors results are consistent with any of these three. If we incorporate the previous evidence for priming, this would favor 3). But I would not put 1) or 2) out of the question. As the authors note, there are other reasons why we may have seen priming in the previous research.

3. A couple of times the article mentions an adaptational argument behind the basis for the priming hypothesis. This isn't clear. E.g., the work by Hills et al 2010 argues that the evolutionary basis for the shared process is likely to be one of exaptation, borrowing an existing spatial search process to search in an interior domain. I think the authors make a novel and more interesting claim but attribute it to Hills et al 2010 when they write: (p4 l 60) "Hills et al. (2010) argue that, with the structure of many search problems hinging on such persist-or-switch decisions, a domain-general mechanism mediating the explore-exploit tradeoff would be adaptive in humans." To be clear, Hills et al do argue that the existence of a search process, that can modulate between exploration and exploitation, is adaptive. But the domain generality is not argued to be adaptive, rather it is informed speculation based on the evidence for shared neuromolecular architecture in comparative biological research across domains and the evidence as explained above about executive function. I wouldn't claim these are adaptive. Indeed, I think a strongly primable search process would be maladaptive, unless we had some reason to believe that tuning the search process for one domain would inform the tuning for a subsequent very different domain. That seems unlikely. Search should be tuned to the environment in which it finds itself. Nonetheless, there may be an adaptive argument here similar to the one proposed in the article and I'm just missing it. If this can be made clearer that would be helpful.

4. I would be curious to hear more about the 50 second warning. I am curious if this 'primes' participants to signal their activity during the anagram task, which they can only do by entering words or moving forward in the task. The instructions say "if you are inactive for a substantial period of time", but I don't think the participants are told for how long, and it also says you'll be financially penalized if you are inactive for longer than the undisclosed amount of time. I'm not sure whether this would cause people to move on from one letter set to another--I suspect it would for some. Obviously, everyone is affected by this equally, but it may nonetheless be a stronger motivation to move on than any cross-domain priming effect. This isn't a criticism: for an online task (or even for a laboratory task) one needs to ensure the participants are fresh and focused. This may have been achieved in the prior studies by only having a short set of pre-test anagrams. Here it is achieved by incentivizing activity.

5. Do people get better at the anagram task as they go on? That is, do they find more words at a faster rate or more words per letter set? Does this settle down before the post-test, suggesting they have an established expectation/strategy? Initially I thought this couldn't be the case,

because the difference is negative for both conditions in Figure 2. But in fact it may be that the first four or five letter sets in the pre-test are enough to find a stable strategy (e.g., "find n words and move on"). Those using such a strategy might show a negative patch time between pre and post, assuming that the first few letter sets took longer, in order to find a reasonable point for switching. I guess I am asking why they are taking less time per letter set after the task. Are they getting better at solving anagrams, calibrating the time per letter set, or simply moving on more quickly, e.g., because they received a warning? Do people who receive a warning change their behavior by moving on more quickly?

6. Because there is no travel time between patches, people will be more inclined to move to the next patch than if the travel time were longer. This is the basic prediction, over many studies, of the marginal value theorem. This isn't necessarily a problem in the present study, assuming that it takes time to warm up to a patch and this time is longer than the time it takes to find the second item after finding the first. Otherwise, the optimal strategy is always to move to the next patch immediately after finding one item. Of course, people may not realize this or act 'optimally' based on some other criterion than rate. I don't think this is a problem here, as the authors exclude people who found less than 20 words, but it might mean that people were strongly incentivized to move on from one patch to the next (in addition to the potential warning for non-activity mentioned above). The authors might want to report the average time to find the first item.

More interesting would be to report the average time to find all items (first, second, third, and so on), average number of items found per patch, and how this changed across patches.

7. On p. 24, l 17, the authors write that "only those who find 20 words in both pre-test and the post-test sessions will be included in the analysis." Does this mean that not all participants found 30 words in the pretest or the posttest? I may have missed it, but I guess this is because if they get through all 14 patches, then they can also complete the anagram task – or is there another reason?

8. 150 seconds on the post-test anagram instructions seems quite long. Did many participants take longer than 20 or so seconds to read these instructions? They had already seen them once before, right? I'm guessing this is a long-tailed distribution. In a laboratory this might not be much of a problem (they might not have phones or other distractions such as email to briefly check), but I am naturally curious if this matters more for an online experiment. Nor am I sure what the right thing to do here is.

In sum, I think this is good and thorough work. Methodologically, it exemplifies good research and a state-of-the-art open science approach. Scientifically, the work raises interesting questions about the role and primability of a common search process for exploration and exploitation across domains.

Reviewer: 2

Comments to the Author(s)

Review of "Priming exploration across domains: Does search in a spatial environment influence search in a cognitive environment?" registered report stage 2 (RSOS-201944.R3), submitted to Royal Society Open Science

Reviewed by Peter M. Todd, Indiana University Cognitive Science Program (original Reviewer 2)

This full version of the paper, reporting the pre-registered and non-pre-registered analyses on the exploration priming study results, is thorough and well-done. It is useful to see the more detailed meta-analyses with Mekern et al.'s data interpreted both in favor of and against the priming hypothesis. The modification of the spatial environments to include multiple patches is helpful for making this study design more readily comparable to that of Hills et al. 2008/2010.

The additional non-pre-registered analyses, enabling consideration of the effect of search behavior (in both the meta-analysis and the current experiment) in addition to the effect of environment structure, are important for trying to understand the differing results between this study and previous research. The authors lay out a range of possible explanations and appropriate strategies for exploring them further. All of this is very beneficial for helping us gain a clearer understanding of search cognition.

One additional measure that could be informative to add to the current non-pre-registered analyses is to see whether environment structure condition differentially affects how people respond after finding or not finding a resource in the spatial search task. Searchers in clumpy environments should stay “local” after finding a resource to increase their chances of finding other resources expected nearby, while searchers in dispersed environments should move away after finding a resource, searching more globally rather than staying local where other resources would not be expected. This was found in Hills et al. 2008 (Figure 2) in terms of turning angle, and in Wilke et al. (2015) in terms of distance jumped in a grid-search task (where win-stay versus win-shift behavior was seen). It could be that the proposed priming effect depends on resource-dependent shifting between local and global search that is common in foraging; if so, then testing for the presence of such shifting in the current data would help further constrain when priming could occur (the test would involve dividing the stepwise clustering measure used on manuscript page 26 into two measures, for those steps following finding a resource versus not finding a resource, and comparing those two measures across the two conditions).

Reference:

Wilke, A., Minich, S., Panis, M., Langen, T.A., Skufca, J.D., & Todd, P.M. (2015). A game of hide and seek: Expectations of clumpy resources influence hiding and searching patterns. *PLoS ONE*, 10(7), e0130976. doi:10.1371/journal.pone.0130976.

An experiment design difference that should be considered in further follow-up studies is the ability to “jump” equally easily anywhere in this grid search task, versus having to move continuously between locations in Hills et al.’s spatial search design. In the current study, local and global search “cost” roughly the same in terms of effort and time, while in Hills et al., global search takes longer to achieve, having to keep moving longer to get further across the grid (though local search takes more actions, in terms of key-presses, to turn and stay in a local area rather than moving more “straight ahead”, away from the previous location, for global search). This difference between the two spatial task versions in terms of distance and travel time may matter for possible priming.

One other difference that may be worth considering further, this time in the results rather than design, is that in Hills et al. 2008 the clumpy spatial environments *increased* the time spent in letter clusters while dispersed spatial environments *decreased* the time spent in letter clusters (Figure 3), but in the current study *both* environment types *decreased* the time spent in letter clusters. Why might that be?

Overall, the data reported are able to test the authors’ hypothesis, which is the same as in the Stage 1 submission; the authors followed the registered experimental procedures aside from a couple of small changes that they received approval for and note in the paper; the exploratory analyses are justified, sound, and informative; and the conclusions are justified given the data.

Detailed comments by page:

Throughout: contractions (e.g. doesn’t, didn’t, we’re, they’d) should be spelled out.

Manuscript page 3 (in lower right-hand corner), lines 26-28: “there may be a domain-general tendency to explore, possibly due to common search processes being involved in different domains (e.g., Hills & Ducas, 2012; Wilke et al., 2009).” -the phrase “domain-general tendency to

explore” here is ambiguous and should be clarified. It could mean that most organisms tend to explore (as opposed to exploit) in most domains they encounter (i.e., exploration is common, across species and across domains). Or it could mean that any given organism will have some individually-specific tendency to explore, from low to high, and that tendency will be similarly manifest (e.g. usually low or usually high) across the various domains that this organism explores in (i.e., individual differences in exploration tendency across domains, as assessed – but not found – by von Helversen et al. 2018 and as explored by the authors in this paper on P27L25-51).

Or it could mean that search itself (including the exploration/exploitation tradeoff) is done in similar ways across species and across domains. The last interpretation seems most likely here, but given the ambiguity, this sentence should probably be reworded to be more specific, e.g. “there may be a domain-general approach to exploration and exploitation across species, possibly due to...” A similar rewording would apply to the same phrase used on P30L7-11 (in two locations).

Reference:

von Helversen, Bettina; Mata, Rui; Samanez-Larkin, Gregory R; Wilke, Andreas (2018). Foraging, exploration, or search? On the (lack of) convergent validity between three behavioral paradigms. *Evolutionary Behavioral Sciences*, 12(3):152-162. DOI: <https://doi.org/10.1037/ebs0000121>

P3L28 and P30L4: “Ducas” should be “Dukas”

P8L39: talking about letter-set timing here, before the cognitive task has been defined, may be confusing – can this be phrased more abstractly, e.g. in terms of patches in the cognitive task? Or describe the tasks briefly before indicating the hypotheses?

P9L53: “averaging this across all 5 grids” – the task has not been described yet, so to make this clearer add e.g. “that each participant searched”

P11L35-47: The participant numbers do not seem to add up properly: $400 - 47 - 2 = 351$, but 347 are indicated; and then there are 174 in the clustered condition and 171 in the dispersed condition, which is 345. And just to confirm, there were nearly twice as many female (221) as male participants (~120)?

P14: Were the participants shown how many words they had generated overall, so they knew how much further they had to go (as was shown for the grid search)? Did they see the words they had already generated from the current letter-set displayed on the screen? A screen-shot of this task as well could be useful.

P23L17: “(i.e., only those who find 20 words” should be “...who find at least 20 words”

P23L56-58: “But to complete the task participants needed to find 30 words and to be included in the analyses they needed to find 20 words.” – this is confusing – what happens if a participant does not complete the task? Do they not get paid? Do they not get to proceed to the next part of the study? Or are there no consequences? And, whatever these consequences of not completing are, their data is still analyzed if they found 20 or more words? And how is “not completing” defined – does it mean that they reach letter-cluster 14 and ask for the next cluster (which presumably does not exist) without having found 30 words?

How many participants did the above classifications apply to? And how many received delay warnings or were removed because of delays or failing comprehension questions?

P27L9: “People in the clustered condition spent fewer minutes in each landscape (M = 5.56,” – “each landscape” should be “these landscapes”, right? That is, people presumably did not spend

5 minutes in each landscape, or else they would be spending 25 minutes on the grid search task overall (and the whole study took ~28 minutes, as indicated on P11).

In the data file, the two environment types are called “rough” and “smooth” – in the Priming Search Codebook file it would be useful to indicate the mapping of these types onto clustered and dispersed.

Author's Response to Decision Letter for (RSOS-201944.R3)

See Appendix D.

Decision letter (RSOS-201944.R4)

Dear Dr Anvari:

It is a pleasure to accept your Stage 2 Registered Report entitled "Priming exploration across domains: Does search in a spatial environment influence search in a cognitive environment?" in its current form for publication in Royal Society Open Science.

You can expect to receive a proof of your article in the near future. Please contact the editorial office (openscience@royalsociety.org) and the production office (openscience_proofs@royalsociety.org) to let us know if you are likely to be away from e-mail contact – if you are going to be away, please nominate a co-author (if available) to manage the proofing process, and ensure they are copied into your email to the journal.

on behalf of Professor Chris Chambers Chris Chambers (Subject Editor)
openscience@royalsociety.org

Appendix A

Dear Professor Chris Chambers,

Thank you for your editorial work on our paper. We have made some revisions to our manuscript and study design and provide responses to all concerns. Below you will find our response to each concern, point by point, with references to the main document where we made changes. We also uploaded a version of the manuscript with track changes so that the editor can more easily see what we have changed: The page numbers in our responses below are with regards to this version of the manuscript.

Editor

Two expert reviewers have now assessed the manuscript and provided very useful reviews. Both sets of comments are positive while also noting a number of detailed methodological and conceptual issues for consideration, including whether the study design and procedures are optimally calibrated to answering the research question. However, these concerns raised appear to be readily addressable through a careful revision and response.

*** Response: We thank the editor and both reviewers for the opportunity to revise our manuscript and for the feedback.*

Reviewer 1

The Mekern et al study cited within as an example of a failure to replicate is really good work. I communicated with Vera Mekern and Bernhard Hommel about the work and some aspects of the design as well as the results. Several differences in the study are quite important, and I think, sadly, confuse the validity of the effect size comparison. First, the Eye-blink rate task is a 6-minute eye fixation task that takes place before and after the spatial foraging task. In Hills et al. 2010, we show that the priming effects we observed only lasts for about 3-minutes, mostly for the first two letter sets. So I wouldn't have expected the priming to survive the 6-minute EBR. Second, participants in Mekern turn more (with a higher turning angle) in the diffuse environment, which isn't what we observed in previous work and indeed would have likely been impossible to manage accurately in the 2008/10 work because the cursor was moving too quickly. I can't explain that difference, though Mekern suggests it may have had to do with the focused-meditation-like nature of the EBR. I really don't know. Possibly that led individuals to persevere on tiny patches in the diffuse task so they then primed themselves to persevere more in the scrabble task. Third, because of this flip in the spatial foraging task results, one could make flipped predictions about what should happen in the scrabble task as well, and of course that isn't helpful, not to mention the 6-minute EBR should have evened things out anyway. So that's all fiddly and complicated and I don't know what to make of it. One thing Vera's excellent work does show, however, is that the EBR is unaffected by the foraging perseveration in the spatial foraging task that precedes it. In this case, it doesn't matter how the spatial foraging task goes, as long as the conditions are different, then one would predict a difference in dopaminergic processes underlying search, if they indeed underly this search. The failure to find a difference in the EBR after the spatial foraging suggests this doesn't work via that mechanism. I am not an expert in EBR, but I trust that Hommel is, so I am inclined to believe that the mechanisms driving our results in 2008/10 may not work as previously

proposed.

*** *Response*: The reviewer raises an interesting and important point regarding the methodological differences between Mekern et al.'s (2017) studies and those of Hills et al. (2008, 2010). As such, we now include these considerations as potential methodological explanations for the differences in the findings between the studies by Mekern et al. (2017) and Hills et al. (2008, 2010). For your convenience we include the excerpts from the manuscript where we have added this information:

“However, Mekern et al.'s (2017) study involved a 6-minute task between the spatial foraging and cognitive search tasks. Given the possibility that priming effects of clustered search can dissipate in short time (see Hills et al., 2010), the contradictory findings may be due to methodological differences. Moreover, the manipulation check in Mekern et al. (2017) showed that participants in the clustered environments had a smaller average turning angle than those in the dispersed environments, opposite to what was found in previous studies (see Hills et al., 2008, 2010). However, even though Mekern et al.'s (2017) manipulation check was in the opposite direction than expected the effect on the main dependent variable was also in the opposite direction to what was hypothesized, so that it could be argued that their findings are in fact consistent with the priming hypothesis.

Although Mekern et al.'s (2017) findings may be explained by methodological differences and argued to be consistent with the priming hypothesis, the meta-analytic effect size calculation for this study produces an estimate of the effect size that is smaller than if we exclude that effect or if we include it as in support of the hypothesis. Hence, our proposed analyses will have additional statistical power to detect effect sizes observed in the Hills et al. (2008, 2010) studies.” (pp. 5-6).

Reviewer: Back to the newly proposed research: There are a few ways in which the proposed paradigm diverges from the original. These are not necessarily criticisms and I think at least one is an improvement and necessary. However, a couple deserve a bit more thought.

1. Participants will be exposed to 14 letter sets before the spatial foraging task. In the original research (Hills, 2008,2010), this was substantially fewer in the pre-test. Arguably, 14 before and after is better for comparison, assuming that something unforeseen doesn't otherwise influence behavior (such as boredom or learning). There is no reason to expect these unforeseen things, so I'm inclined to think this is in an improvement.

*** *Response*: The reviewer's assessment matches our own and we thank the reviewer for the kind words.

2. Participants are able to submit three-letter words. The original uses four-letter words. I worry that three-letter words will allow participants to find solutions with less effort. This is a problem if one believes that the effortful part of the task is the part being primed, which I do. I vaguely recall spending some time on finding letter sets that provided enough four-

letter solutions. Should this really matter? I'm not sure. In semantic memory search tasks, like category fluency, there is evidence of an executive search effect (e.g., Rosen & Engle, 1997). One would only expect to see this effect during effortful components of a search. If the search task is too easy, then it's not clear what to expect.

*** *Response:* We thank the reviewer for this comment and suggestion and have made some changes. First, we searched and put together 29 letter-sets with an average of 12.7 (4.5) solutions, which we judged to be close enough to the average number of solutions to the original studies (see p. 13). Second, we made sure that the 14 letter-sets in the pre-test phase closely matched the 14 letter-sets in the post-test phase, particularly for the earlier letter-sets, in the number of solutions that could be generated from them. Third, we changed the instructions to reflect the fact that participants are now required to use the letter-sets to make words made up of at least 4 letters. Finally, all participants receive the letter-sets in the same order because we have organised them in such a way that each letter-set in the pre-test session is matched to a letter-set in the post-test session in the corresponding position such that both have a similar number of solutions. For example, letter-set 1 in the pre-test session has 14 solutions, as does letter-set 1 in the post-test session. To reflect these changes, we have made the following changes to the manuscript:

“The 14 letter-sets for the pre-test anagram task will be the same for all participants, presented in the same order. [...] The 14 letter-sets for the post-test anagram task will be the same for all participants, presented in the same order.” (pp. 12-13).

“We used R to randomly produce letter-sets made up of 6 letters, four consonants and two vowels, and we continued to generate letter-sets until we had a sufficient number of them with a similar number of average solutions to Hills et al.’s (2008, 2010) letter-sets” (p. 13).

“The pre-test letter-sets have a similar number of solutions compared to the post-test letter-sets in the corresponding position (e.g., letter-set 1 in both the pre-test and post-test sessions has 14 solutions)” (p. 14).

“The letter-set for the practice round is the same for every participant (i.e., “Y E A W H S”)” (pp. 15-16).

3. The proposed paradigm uses a 0-second time cost between letter sets. The original used a 15-second waiting period. The reason for this in the original research is because the task is designed to represent a patch-foraging task, similar to the spatial foraging task, and also similar to the marginal value theorem. According to the marginal value theorem, if there is no travel time between patches, and patch resource intake rates are always monotonically declining after arrival, then foragers should move between patches instantly. Obviously that's in the mathematical limit of no-travel costs whatsoever, which is unlikely to exactly match the anagrams. It takes time to read the letters and start thinking about how they could be manipulated. However, a 0-second waiting time invites participants to give up as soon as they encounter any difficulty, as the next patch will be a rich source of new solutions. Alongside the 3-letter words, I think this is likely to make the task too easy. Of course, it may not, but I feel it's better for me to point this out even if I'm wrong about how the task will work in practice. I guess I would ask how long it takes participants to find 30 items and how

quickly do they switch between letter sets? If this comparable to the earlier work, then it is a better replication.

*** *Response:* We agree with the reviewer that including a waiting period between letter-sets makes the anagram task more similar to patch-foraging task. However, given that the spatial task we are using is not a patch-foraging task we think this is less of an issue for the present study. Moreover, participants who move too fast and get through the 14 lettersets without finding the required number of words will be excluded so again this shouldn't be a problem. As reviewer 2 notes (see "Detailed comments" section, under P.14), including a zero time-cost to switch lettersets is reasonable for the design we have proposed. Moreover, when participants start finding words, they may feel on a roll, given that their minds will have adjusted to these words and finding one word can facilitate the finding of the next word. Participants will therefore likely continue as the initial difficulty of finding the first word has already been overcome until the patch is more exhausted (i.e., the words are running out), at which point when they can switch to the next anagram (as they would switch to a new patch). Perhaps most importantly, the hypothesis states that people searching a spatial environment with clustered resources will spend more time in each letterset; so even if in our task people generally spend less time in each letterset, we should still observe participants in the clustered condition spending relatively more time in each letterset than participants in the dispersed condition, if the hypothesis were true. We will report in our analyses how long participants take in each letterset and how quickly they switch, as this is the main test of the hypothesis. Nevertheless, the priming hypothesis does not make stipulations about how long participant should be in each letterset. It is a very general hypothesis, coming from quite a general theory, stating that how resources are distributed in a spatial environment that people are searching will impact their search behaviour in a cognitive search task. Hence, if our findings do not find support for the hypothesis it may be that the hypothesis should be adjusted to be less general and more specific about when it is expected to be supported.

4. The grid search is the biggest difference from the prior work. I think this is an improvement. As the authors note, this tests not so much a replication as a test of the generalizability of the results to a very similar situation. It also potentially overcomes the arousal problem. Moreover, it avoids the problem observed in Mekern, that the spatial foraging manipulation didn't replicate.

*** *Response:* Thank you and we agree.

5. In the analysis, will the analysis include a look within individual letter sets, as done in 2010?

*** *Response:* We will conduct exploratory analyses. However, we have not decided on all of the exploratory analyses which we will conduct yet. We believe that the analyses we are pre-registering are the main analyses that provide the most conclusive and confirmatory test of the hypothesis. In registered reports we are required to only include the confirmatory analyses, and even prevented from including exploratory analyses in the Stage 1 manuscript. Nevertheless, we have considered looking within individual lettersets and indeed we may conduct such analyses exploratorily.

6. Also, will the analysis include a focus on individual differences between the two tasks, following the last paragraph of the results in Hills et al., 2008? Neither 5. or 6. were done in Mekern's work, but I think they would have added value. Even if the manipulation failed to work as expected, the replication of a relationship between foraging externally and internally was still available for testing.

*** *Response:* As per our response to the previous point, we have not included exploratory analyses in the Stage 1 manuscript. Nevertheless, this particular exploratory analysis is certainly one that we will conduct. Specifically, we will examine the correlation between how clustered/dispersed people search in the spatial environment with how much time they spend in each letter set. A significant negative correlation would suggest that people who search in a more dispersed way in the spatial environment also search in a more dispersed way (i.e., switching between letter sets more quickly) in the anagram task. Importantly, the priming effect is the confirmatory test that the proposed study is focused upon and so it is that test which we include in the Stage 1 manuscript. We will, of course, conduct and report exploratory analyses that can also speak to whether there is a domain-general search process, such as the correlation described above.

Overall, I think the design is a close replication with some minor variations and would be interesting and worthwhile. Plus the authors provide confidence that the work will be well done. The authors should consider whether they think the comments above warrant changing the design to limit solutions to only four or more letters and to increase the wait between new letter sets, as well as whether the final two comments (5 and 6) would further help address the underlying research question about a domain general search process.

*** *Response:* We once again thank the reviewer for their input and valuable feedback. As per our responses above, we have opted to use different letter sets to make the anagram task more difficult by requiring participants to find 4-letter words, but to keep the zero switching time between letter sets. Importantly, we will conduct exploratory analyses that may also speak to whether there is evidence for a domain-general search process but for the confirmatory test we are most interested in the priming effect which we think is the evidence from past research that provides the strongest and most stringent test of the domain-general priming hypothesis.

Reviewer 2

Search for resources of different types is a crucial adaptive behavior for all mobile organisms, and there is ongoing debate about whether identical, similar, or different cognitive mechanisms are used for search in different domains. Priming studies between different search tasks have been used to inform this debate, but the results have been mixed, so a high-powered pre-registered replication would go far to advance this discussion and the understanding of search cognition. This proposed study design looks to be a good way to

replicate and test the generality of earlier search priming study results, including of Hills, Todd, and Goldstone (2008, 2010) and Mekern (2017). The authors make good arguments for the importance of replicating the earlier studies. Their new design decisions are also appropriate, including increasing the power with a larger sample of participants, using a number of exclusion tests to ensure consistent sustained attention of the online participants, and having a fixed number of possible search actions per landscape to remove one possible alternative explanation of search priming results in terms of key-press based differences in arousal between the conditions. The methods are described in a clear and detailed manner that will allow others to readily replicate this study. The results of this study will be very useful in assessing the evidence for priming between different search domains, and consequently for pushing the field of search cognition forward.

**** Response: We thank the reviewer for the kind words and for the solid summary of our proposed study.*

There are though a few differences in this design from past study designs that should be considered if the results of this replication study are inconclusive. These are not arguments for changing the proposed design (aside from adding some additional measures and analyses, mentioned below in the detailed comments by page), but rather some further ideas about what could lead to priming between search tasks, if/when it does exist.

First, in the Hills et al. and Mekern spatial navigation search, and in the anagram task, it takes longer to make "bigger" steps (over the landscape, or to a very different word), but the current clicking search task is more like visual search, where longer steps across the landscape do not take (much) longer to produce (participants just have to move the mouse). Does that difference in the spatial task versions break down the connections between the two search tasks in some way?

**** Response: One option in our design decisions was to include a step-time to switch between lettersets in the anagram task as well as to use a grid-search task in which participants could only search the grid by clicking on one of the options adjacent to the option previously selected. However, the grid search task we are using is a very close adaptation of the extremely widely used multi-armed bandit task typically used to model exploration-exploitation dilemmas. As such, we decided to remain true to the multi-armed bandit design which includes no time to travel from one option to another. We believe that this should not be problematic for a test of the priming hypothesis, given that the hypothesis is very general—i.e., the hypothesis is that search in a spatial environment will prime search in a cognitive environment. Therefore, from a theoretical perspective, have zero time to switch between lettersets and zero time to move from one option to another option anywhere in the spatial environment should not be problematic.*

Second, the fact that the current landscapes only have one big patch means there is no "exploit-to-explore" (patch-leaving) decisions to be made, which is different from the previous versions where there were multiple patches and so multiple switch-patch decisions. The anagram word puzzle includes multiple exploit-to-explore decisions, and in fact that is effectively the main thing being measured. How will the single patch landscapes

change priming and behavior? (An alternative design would be to still have one patch, but of varying size across different landscapes, and let participants search as long as they want and switch to the next landscape whenever they want—this would invoke the exploit-to-explore decisions in the Hills et al. and Mekern designs, and in the anagram task.)

*** *Response:* The reviewer raises an interesting point about a difference between the original and current study designs: whereas in the original studies participants could move from one resource patch in the to another within the same spatial environment and thereby switch from exploiting one patch to exploring (or searching) for another patch, in our proposed design each spatial environment (i.e., grid) has only one patch for those in the clustered condition. The reviewer seems to argue that in the proposed study the spatial search task is thus less similar to the anagram task. However, we would argue that our proposed design provides a much stronger manipulation. The priming hypothesis essentially states that how resources are distributed in a spatial environment, clustered versus dispersed, will prime people's search in a cognitive environment such that they will engage in more clustered versus dispersed search, respectively, in the anagram task. Because in our design the clustered condition involves grids with one large patch, participants should be more strongly primed to continue exploiting the one patch, or, put another way, to explore in a more clustered way, than if we had grids with multiple patches. In contrast, participants in the dispersed condition will search grids in which the resources are dispersed such that there is not a single patch of resources in the spatial environment. Hence, people in our clustered condition should be strongly primed to search in a clustered way (sticking within a resource patch once they find one) whereas those in the dispersed environment should be strongly primed to search more widely (switching from one patch to another in search of resources that are dispersed). Taken together, rather than it being problematic, our design provides a greater likelihood that the priming effect will be observed if it is a true effect. If we find no evidence for the priming hypothesis, then the hypothesis may be argued to be too general and either the hypothesis or theory should be revised to accommodate the findings.

Third, because participants in the clustered landscape of the current design only explore one big patch continuously (once they find it) and get a near-constant stream of reward, they might therefore think the world is richer overall than do participants in the diffuse environment, because they have seen a much higher rate of return while searching. Would that different estimate of the "richness" of the world between the patchy and diffuse conditions affect behavior in the anagram task? If so, that could possibly mask any effect of the structure of the environment (clustered vs. diffuse). It could be that participants in the clustered environment would be primed to think the world is very rich, e.g. full of big patches, which could lead them to then stay longer in each patch in the anagram task. Or possibly if they are primed to think the world is rich, then when they are in each anagram patch and they feel themselves slowing down in producing more words, they may think, "this *isn't* one of those big rich patches, so I should leave it right away and look for a big patch!" --leading to the opposite behavior, namely leaving sooner. Either way, if the single-cluster manipulation in the proposed study changes how rich the spatial world feels, that could make a difference in the results for an unintended reason.

*** *Response*: The reviewer raises another interesting point but which we think could be equally applied to the original designs. For instance, in the original studies the clustered environments had larger resource patches than the dispersed environments. So people who searched clustered environments would have gotten a more constant stream of rewards than those in the dispersed environments. In fact, this was the case (see Hills et al., 2008; Mekern et al., 2017). In Hills et al. (2008, 2010) the number of resources found in the spatial environments was not a significant predictor of time spent in lettersets and that including resources found as a covariate did not change the results. This evidence suggests that the number of rewards or resources found should not impact the time spent in lettersets nor is it likely to be an explanation of the priming effect. Therefore, in conjunction with the reasons provided in our response to the previous point (i.e., the current design provides a much stronger manipulation of clustered search), we argue that the proposed design with clustered grids involving only a single patch as compared to dispersed grids involving dispersedly distributed resources is a good design choice. In exploratory analyses we will also examine whether number of resources found is a significant predictor of time spent in lettersets and whether adding number of resources found as a covariate changes the results of the main test, as was done in Hills et al. (2008, 2010).

Detailed comments by manuscript page number (e.g. “Page X of 28”)

Pp. 4-5: “Hills et al. (2010) argue that, with the structure of many search problems hinging on such persist-or-switch decisions, a domain-general mechanism mediating the explore-exploit tradeoff would be adaptive in humans”—this is not the only possibility—perhaps more likely is that a mechanism that originally evolved to solve the explore-exploit tradeoff in one domain (initially probably spatial) could be “exapted” in the course of evolution and used subsequently in a variety of other domains (e.g., in internal search).

*** *Response*: We thank the reviewer for this additional theoretical argument which was also made by Hills et al. (2010). We have thus included it in the manuscript. For your convenience we include the excerpt here:

“Hills et al. (2010) further argued that a mechanism originally developed and used to deal with the explore-exploit tradeoff in one domain (likely to be spatial search) could have been duplicated through evolution and used in other domains (e.g., internal or cognitive search).” (p. 4).

P6: The Mekern (2017) study results *do* appear to fit with the priming idea, in that more exploitative local search in the spatial task—staying near where rewards have been found earlier, in terms of greater turning angle after finding rewards—did correspond to longer search time in subsequent word puzzle patches. The difference from the Hills et al. 2008 study is that the Mekern spatial environments were constructed in a way that made the *diffuse* environments elicit more local search than the *clustered* environments. So the current authors’ meta-analysis here should perhaps be redone by comparing results across the studies on the basis of the local versus global search *behavior* shown in the priming (spatial foraging) task, rather than the type of environment (patchy vs. diffuse) used in the

priming task, which may not (as in Mekern's case) elicit the expected type of behavior.

In contrast to the Mekern study, the spatial environments in the current paper are constructed in a way that will probably maintain the same relationship between spatial structure (patchy vs. diffuse) and heightened local search (more in the patchy environments than in the diffuse ones) seen in the Hills et al. 2008 design (this is because the diffuse environments in the current design, similar to Hills et al., usually have only one rewarded location alone, rather than a small cluster). As a consequence, the analyses, which should be done in terms of both constructed spatial structure and observed amount of local search, will probably both point in the same direction with respect to the presence or absence of priming between the two search domains.

*** *Response:* We thank the reviewer for this alternative explanation of the Mekern et al. findings. As can be seen in our response to reviewer 1 (in the first paragraph) we have added this alternative explanation on pp. 5-6 of the manuscript. Nonetheless, as per our response to reviewer 1 (first paragraph), including the effect size observed in Mekern et al. (2017) results in a smaller meta-analytic effect size estimate for which we power our analyses. Hence, we are using a more conservative approach to arrive at a study design and analysis plan with higher statistical power to detect smaller effect sizes. Moreover, although the alternative explanation from the reviewer is interesting, the original hypothesis and confirmatory test in Mekern et al. (which was not supported) was the same as in Hills et al. (2008, 2010). As the reviewer notes, and as reviewer 1 also noted, Mekern et al.'s manipulation failed (or at least, it went in the opposite direction to what was intended). Finally, the alternative explanation of Mekern et al.'s findings is not likely to impact the proposed study. Our design, as the reviewer notes and as shown in the pilot, has a very strong effect on the manipulation check in the expected direction, and so we expect the direction of results to be the same as in Hills et al. (2008, 2010).

P10: Why was a measure of dispersion of searching across the *entire* search (all 50 tiles) used, rather than a measure of stepwise dispersion (from one selected tile to the next), as in the measures from Hills et al. and Mekern of turn angle from one step to the next? The global mean used here could obscure search patterns at the local level—for instance, the obtained value of $M=6.63$ in the clustered condition does not tell us anything directly about how far a participant moved from one tile to the next when searching (where presumably that distance is closer to 1-2). It is important to also measure the step-to-step clustering or dispersion (e.g. akin to turn angle or win-stay-lose-shift behavior) to assess whether the task and environment structure are successfully eliciting clustered versus diffuse search behavior in the participants, and hence are setting up the appropriate differences for the priming effect to be measured (as discussed above regarding the Mekern study where this appropriate elicitation did not happen). A related measure of local search behavior is used with a similar clicking grid search task in this paper:

Wilke, A., Minich, S., Panis, M., Langen, T.A., Skufca, J.D., & Todd, P.M. (2015). A game of hide and seek: Expectations of clumpy resources influence hiding and searching

patterns. PLoS ONE, 10(7), e0130976. doi:10.1371/journal.pone.0130976.

*** *Response:* The measure the reviewer suggests is a theoretically good one, but one which is better suited for foraging tasks, such as in the original studies, where participants can't skip and go back to different sections of a resource patch. In our grid search task, participants may find a patch, pursue the resources in the patch in one direction until they reach the boundary of that patch, and then jump several options backwards to continue their exploitation of the patch on the other side of where they first found the patch. In the step-wise analysis of clustered search, this jump so described would be considered non-local search or as jumping to another patch. However, conceptually, the participant would still be searching the same patch, just a different section of it. (See Figure below for a poorly drawn illustration.) Therefore, a better measure of clustered search in the current design is the one we have proposed and used in the pilot study and which we propose to keep for the main study. Nevertheless, results from our pilot study show that the manipulation has a strong effect on the measure of step-wise clustered search: people in the clustered condition ($M = 2.45$, $SD = 0.98$) searched options more locally than people in the dispersed condition ($M = 5.32$, $SD = 2.25$), $t(42.7) = 6.37$, $p < .001$, $d = 1.65$ (see R code lines 101-110). However, for the reasons given, we think that the measure we have used and submitted in the manuscript is a better measure of clustered search in our study and thus we opt to keep it.

Imagine someone starts at the option in the blue circle in the center of the patch and they click on the options next to each other one at a time following the blue line. Once they reach the edge of the patch (the poorly drawn blue arrowhead), they then jump back to the other side of where they had started (follow the green line) and click on an option next to the first option they found in the patch. In the step-wise clustered search calculation, this jump would be considered as non-local search, or as leaving the patch. However, it is clearly a situation of continuing search within the same patch. Our measure of clustered search captures this whereas the step-wise measure does not.

P14: This seems like a reasonable design choice to not include a switching cost (time delay)

between anagram “patches” (perhaps a downside could be some participants leaving all the patches too quickly and running out of patches, but that seems unlikely).

*** *Response:* The reviewer’s assessment matches our own. Moreover, people who run out of lettersets without finding the required number of words will not be included in the analyses.

P17: Is there a reason for not having exactly 80 reward tiles per grid, so the overall available award is exactly equal across landscapes?

*** *Response:* The randomization procedure of creating the grids produced an uneven number of tiles across grids. In our assessment, this should not be problematic for testing the priming hypothesis. There are always more rewards than there are clicks for participants, and the important factor is whether the environment involves resources clustered together or dispersed apart. Moreover, on average, the grids in the different environments have similar numbers of rewards. We have now added the following to the manuscript to clarify:

“In the clustered landscapes, the average points per tile is $M_{clustered} = 19.94$ ($SD_{clustered} = 0.54$) and there are between 77 and 83 reward tiles in each landscape; in the dispersed landscapes, the average points per tile is $M_{dispersed} = 20$ ($SD_{dispersed} = 0$) and there are exactly 80 reward tiles in each landscape. The difference depends upon the algorithm used to populate the landscapes” (p. 17).

P18L52: “participants have 50 trials in which they can select any tile and gather points” – since the number of clicks in each landscape search is limited, rather than the time per search as in Hills et al. and Mekern, it would be good to also measure the time in seconds taken per landscape to see if that varies across environment types as a possible predictor of priming.

*** *Response:* We agree with the reviewer and already intend on doing this, as we have done and reported for the pilot study: “Participants took an average of 7.7 (3.0) minutes to complete the grid search task: 6.7 (2.1) minutes for clustered grids and 8.4 (3.5) minutes for dispersed grids” (p. 10). We will use this in exploratory analyses.

P18L42-45: “clustered landscapes consist of only one patch of resources so that participants are likely to continue exploiting. This will make the exploitation prime in our study more pronounced.” –As discussed above, while having only one patch will increase the exploitation time compared to Hills et al.’s design, it will remove the need to monitor success and decide when to switch (leave a patch and explore for a new one). So different results in this study could possibly be attributed to lack of switching experience in the clustered landscapes.

*** *Response:* As per our response to the reviewer’s second and third points, we do not believe there is a plausible explanation that could not be equally applied to the original designs. Hence, if we observed different results in the present study, then either the

hypothesis or theory need to be revised to be more constrained and less general; the current hypothesis and theory are quite general and make no claims about the specifics of the environments other than whether resources are clustered or dispersed. Moreover, in exploratory analyses we will examine whether the number of resources found could explain the priming effects, as done in Hills et al. (2008, 2010).

P19L26-30: “The points that you can earn in each tile are from 0 to 100. Revealed tiles are also colour coded, as a visual aid to help you in this task. Darker colours correspond to larger rewards.” –why talk about the points per tile as though there is a range of values, when there is actually only either 0 or 100 points per tile? It seems unnecessary to lead people to expect something that is not actually the case (i.e., that there are some intermediate reward values).

*** *Response:* The reviewer raises a very good point and we have made adjustments to address this. The participants are now instructed that “The points that you can earn in each tile are either 0 or 100” (see p. 19).

Appendix B

Dear Professor Chris Chambers,

Thanks again for your editorial work on our manuscript and the opportunity to address the continuing concerns of the reviewers. We've addressed these concerns and respond to each point, in detail, below. We hope we've understood the reviewers' points correctly and responded sufficiently well.

Kind regards,

Authorship team.

Editor

Both reviewers from the previous round returned to assess the revised Stage 1 manuscript. Both now recommend IPA following some remaining minor revisions to address conceptual concerns and potential design limitations. Any points regarding exploratory unregistered analyses can be addressed at Stage 2. If the authors are able to address these remaining issues in a final Stage 1 revision then IPA will be forthcoming without requiring further in-depth review.

** Response: Thanks again and we hope that we've successfully addressed the remaining issues. See our point-by-point responses to the reviewers' concerns below.

Reviewer: 1

Overall, I appreciate the authors attention to detail and response to the previous comments. I think the proposed study will test a general form of the hypothesis that 'search in one domain influences search in another domain', which the authors claim is their primary goal.

They are also testing hypotheses about how that search process might work. For example, one question is whether or not the mechanism is meta-cognitive: people realize they are searching more locally or more globally and then unwittingly apply this 'frame' in a new domain. Or whether it is more procedural: Some low level 'turning/look-back' process is activated and applied across domains. Moreover, the adjustment in turning rate may be a function of how far the individual has to travel between patches or how long it takes. If it's more meta-cognitive, then knowing that letter sets have no travel time between them may be important, as this may be integrated into the overall strategy. If the authors have found that people still spend sufficient time in letter sets with no travel time between them, then this shouldn't be a problem. If it's a low-level turning process that is calibrated (e.g., like it might be for the Margial Value Theorem, see Adler and Kotar), then the time to move from one place to another in the spatial landscape may be important. It will be interesting to know that and this can possibly be tested in the exploratory analysis as well.

I agree with the authors that a very general form of the hypothesis is not based on these

subtleties about the underlying process.

**** Response:** We thank the reviewer and agree that we are interested in testing the general form of the hypothesis. The reason we didn't add travel times was to keep the two tasks more similar: i.e., there is no enforced travel time between "patches" in the grid search task and so we didn't include travel time in the letterset task. We aren't quite sure about whether the reviewer's comment suggests that we should add a time-delay between lettersets or not. Given that the reviewer's final statement regarding this issue states that the general form of the hypothesis is not based on the subtleties raised in the larger paragraph preceding it, we think that the reviewer is suggesting that there's no need to introduce a travel time between lettersets and that, instead, we should examine those subtleties in exploratory analyses. If we've misunderstood, we're happy to add a travel time between lettersets.

Minor comment: I wouldn't say 'duplicated' as on p. 16 Line 13. This has a very specific biological/genetic meaning, which often means it can act independently from the copy on which it is based. We're talking about behaviour here, so the semantic mapping is perhaps a bit looser, but 'exapted' is probably the word the authors want and the one truer to the original Hills et al hypothesis. Exapted simply means the process was adapted to a new purpose. In fact, I've often felt it is the same purpose merely confused by the fact that people tend to think there's a difference between external information and internal information. There is a difference, but it may not be one that is meaningful to our search processes, however they may work.

**** Response:** We've changed "duplicated" to "exapted".

Reviewer: 2

The authors have done a thorough job of responding to the two reviewers' comments and have argued well for their experimental design. I have only a few remaining comments where the authors can consider further explanatory possibilities and exploratory measures and analyses, in response to the indicated locations in the revised manuscript. I look forward to seeing the results of this replication and continuing the discussion.

P7L22-27: "Because in our design the clustered condition involves grids with one large patch, participants should be more strongly primed to continue exploiting the one patch, or, put another way, to explore in a more clustered way, than if we had grids with multiple patches." – I disagree with the authors' argument here. Search in a clustered environment can lead to later behavioral differences (i.e., can have a priming effect) compared to search in a dispersed environment only if those two searches are different; and what can make those two searches different is having to deal with clusters in the former and not in the latter—for instance by finding clusters, staying in them, and leaving them when it's appropriate to do so. In the current design with one big cluster, there is little finding to be done, relatively

little need to do anything to stay inside the big cluster, and no leaving decisions to be made—making the appropriate behaviors here not so different from those for the dispersed environment. (If the single cluster size were increased, these differences would decline toward zero.) So this does not seem to me to be a design where priming will be stronger, but rather weaker. The question is whether the low levels of these differences will be sufficient to induce observable levels of priming (which will be interesting), but if no priming is found, it may be a consequence of not using environments of the sort where such priming should be expected.

** Response: We thank the reviewer for the kind words and take the reviewer’s point. Therefore, we’ve created a new set of clustered environments in which there are 3-5 patches of rewarding tiles. We therefore changed the grid search task so that participants have 80 clicks in each grid across a set of 5 grids, to keep the time spent on the grid task similar to the previous version and to ensure that there are sufficient clicks for participants to travel between patches. In the new task, given that participants have 80 trials per block with an average of 80 rewarding tiles per grid split into 3-5 patches, we think that there is a sufficiently high likelihood that any single patch will become depleted enough that participants will need to search for and find new patches to exploit. We have updated the manuscript to reflect these changes to the grid search task, including instructions to participants (see pp. 17-21): we’ve taken out excerpts from important parts and provide them below for your convenience. We’ve also updated the OSF project page with the new clustered landscapes. Here is an example of a clustered grid with patches (this is also used in Figure 1 in the manuscript):

As before, “The 40 grids for each of the two environment types had extremely similar average points per tile: In the clustered landscapes, the average points per tile is $M_{clustered} = 20.13$ ($SD_{clustered} = 0.49$) and, as noted earlier, there are between 77 and 83 reward tiles in each landscape; in the dispersed landscapes, the average points per tile is $M_{dispersed} = 20$ ($SD_{dispersed} = 0$) and there are exactly 80 reward tiles in each landscape.” (pp. 17-18).

Importantly, we also ran another pilot test to make sure that the manipulation still had a sufficiently strong effect on the manipulation check. We have therefore updated the pilot study section of the manuscript (see pp. 9-10). Anticipating that the effect size would be smaller than before, due to the clustered grids now having resources distributed further apart (i.e., in 3-5 patches) than when the clustered grids had a single patch, we aimed for a total sample of 100 participants. Due to a software error, we lost data from 14 participants, leaving a total of 86 participants for the pilot.

“Our pilot test showed that the manipulation had a significant and strong effect on the measure of how clustered/dispersed people searched. Participants in the clustered condition ($M = 10.07$, $SD = 2.20$) searched in a more clustered way than participants in the dispersed condition ($M = 12.15$, $SD = 2.07$), and this difference was very large, $p < .001$, $d = 1.40$. Participants took an average of 7.3 (2.7) minutes to complete the grid search task: 6.7 (1.9) minutes for clustered grids and 7.8 (3.3) minutes for dispersed grids ($p = .107$). The data and analysis code can be found on the OSF project page (see link below). We conclude that the manipulation is sufficiently strong for the purposes of the proposed study.” (p. 10).

P9: With regard to Mekern’s results, the hypothesis is *not* that “search in a spatial environment will prime search in a cognitive environment” – it’s that the *type* of search that an organism does in a spatial environment will prime the same type of search in a cognitive environment, and that is also what Mekern’s results show, when analyzed in terms of the type of search, not the type of environment. So the suggestion is to do the analysis of this new proposed study, and the meta-analysis of all the studies being compared, on that basis, of search type, rather than of environment type.

** Response: We accept the reviewer’s point, and so we’ve updated the relevant section of the manuscript and meta-analytic effect size estimate. Page 6 now says:

“To assess the meta-analytic effect size estimate from the three studies (i.e., Hills et al., 2008, 2010; and Mekern et al., 2017) combined, we used Goh et al.’s (2016) calculator, inputting the relevant information (reported effect size estimate, group sample sizes, means, standard deviations, and/or the t statistics). We calculated two fixed-effects meta-analytic effect size estimates. First, we calculated the meta-analytic effect size with Mekern et al.’s findings interpreted as being in favour of the priming hypothesis, for the reasons provided in the preceding paragraph. Second, we calculated the meta-analytic effect size with Mekern et al.’s findings interpreted as being against the priming hypothesis. When Mekern et al.’s (2017) effect size was input as consistent with the priming hypothesis, the meta-analytic effect size estimate was statistically significantly in favour of the priming hypothesis (mean Cohen’s d of 0.64, $CI_{95\%}$ [0.32, 0.96]); when input as inconsistent with the priming hypothesis the meta-analytic effect size estimate was statistically *nonsignificant* (mean Cohen’s d of 0.26, $CI_{95\%}$ [-0.06, 0.58]). (See Supplementary Materials for details of parameters obtained from each study for the meta-analysis, and the Open Science Framework, OSF, project page for the calculation of the meta-analytic effect size in the excel spreadsheet, Tabs 1 and 2; <https://osf.io/fmax6/>; check “READ ME” file first.)

In addition, we added a condition to our hypothesis such that if people in the dispersed condition engage in more clustered search than people in the clustered condition as will be evident in the manipulation check, then the hypothesis should be reversed. Page 8 now says:

“Specifically, we predict that people who engage in a spatial search task in which the resources are clustered, compared to those for whom the resources are dispersed, will spend more time in each letter-set of the cognitive search task, as measured by mean time spent in each letter-set post-test minus mean time spent in each letter-set pre-test. Note, however, that if people in the dispersed environments engage in more clustered search than people in the clustered environments, as will be shown on the manipulation check, then our hypothesis will be reversed such that people in the dispersed condition should spend more time in each letter-set than people in the clustered condition.”

P10L6ff: On the dispersion measures to use, I appreciate the authors’ argument that the local dispersion measure could count some instances where the searcher jumped to another part of the same patch as being non-local – but the usefulness of that measure would be to assess local versus non-local movement, not necessarily within-patch versus not-within-patch movement, and this would still be useful to know for the reasons spelled out in my previous review regarding distinguishing different types of search behavior. Hence reporting both measures (even if one is exploratory) would help researchers to understand better what people are doing in the different search environments.

** Response: In the Stage 2 manuscript we will also report the additional measure that the reviewer has noted as part of exploratory analyses.

For now, it suffices to say that our manipulation once again had a strong effect on the measure of step-wise clustered search (the alternative that the reviewer suggests): People in the clustered condition ($M = 2.53$, $SD = 0.67$) searched options in a more clustered way than people in the dispersed condition ($M = 3.96$, $SD = 1.70$), $t(48.7) = 6.04$, $p < .001$, $d = 1.30$ (see R code lines 136-151).

P11L41ff: It will be interesting to look (exploratorily) at the time taken per grid to see if that predicts the amount of priming found at the individual level in the different environment types, and also whether it is correlated with the dispersion for each grid—does it take more time (and maybe more thought) to search dispersed environments than clustered ones? Do people who search clustered environments in a more dispersed manner take longer and show less priming of search in the anagram task?

** Response: We aren’t completely sure of how we would run these analyses, but given their exploratory nature (and of course they are interesting questions to consider) we will examine them more closely and the best we can at Stage 2 of the manuscript.

Appendix C

Dear Editor,

Please accept our manuscript, “Priming exploration across domains: Does search in a spatial environment influence search in a cognitive environment?”, as a Stage 2 Registered Report at *Royal Society Open Science*. Note that I’ve left the statement about submission processes in case this is still required (see below). I Stage 1 manuscript was previously accepted, and we are now submitting the Stage 2 manuscript. The Stage 1 manuscript was pre-registered on the OSF: <https://osf.io/jge9p>. We changed the tense of the manuscript when describing the study to be in past tense. In addition, apart from a pre-approved alteration, there were some very minor alterations to the manuscript after it was pre-registered which we note now.

Important Notes

After we received IPA for the Stage 1 manuscript we preregistered the manuscript on the OSF: <https://osf.io/2gu46>. However, before starting data collection we realized that one inclusion/exclusion criterion was not yet updated following the review procedure. We therefore contacted the journal and they gave us approval to make this change and to put a note to the editor in the cover letter about this change. Specifically, criterion 3 required that participants complete the anagram tasks (i.e., find a total of 30 words) to be included in the analyses. This needed to be changed to finding a minimum of 20 words since 30 words was a high requirement after a change in the Stage 1 review process. The updated pre-registration then said that “Participants who do not complete the anagram tasks will be excluded from the analyses (i.e., only those who find 20 words in both the pre-test and the post-test sessions will be included in the analyses)”. See the updated preregistration here: <https://osf.io/jge9p>. But to complete the task participants needed to find 30 words and to be included in the analyses they needed to find 20 words. We’ve therefore changed this in the manuscript to reflect the fact that participants who didn’t find the minimum of 20 words were excluded from the analyses, as opposed to those who didn’t complete the task. To make this clear in the manuscript, we’ve added a footnote (see footnote 4, pp. 22-23).

In addition, we preregistered a base payment of £2.00 pounds but, before collecting data, we deemed this insufficient and increased the base payment to £2.60. This is noted in brackets on p. 11 of the manuscript.

Finally, in the instructions for the grid search task, the pre-registered manuscript tells participants that “previously revealed tiles cannot be reselected”. We changed this to “**Each tile can be selected only 1 time**” to improve clarity. We’ve made a footnote of this change on p. 20 of the manuscript (footnote 3).

Summary of the Paper

Research and theorizing suggest that there may be a generalized cognitive mechanism that connects exploratory behaviour across domains. Specifically, two published studies (Hills et al., 2008, 2010; cited in the manuscript) suggest that experience searching a computerized spatial environment influences subsequent search patterns in a cognitive environment (i.e., an anagram task), supporting the cross-domain priming hypothesis. The cross-domain priming hypothesis serves, at least partially, as the foundation of an important theory about a general cognitive mechanism that controls exploration-exploitation behaviour across all domains.

However, the cross-domain priming hypothesis needs further empirical verification for three main reasons that are extensively discussed in the introductory section of the manuscript—namely, the existence of mixed experimental evidence, to assess the generalizability of previous results to a similar task, and reducing confounds in previous experimental designs).

Thus, we proposed a large pre-registered test of the cross-domain priming hypothesis. Our proposal was accepted as Stage 1 and we are now submitting a Stage 2 manuscript.

The pre-registered hypothesis test was not supported. An equivalence test showed that the effect was smaller than the smallest effect size of interest. We conducted several exploratory analyses. One out of all of these showed some consistency with past findings.

Failing to observe a cross-domain priming effect, we conclude that evidence for the generalizability of the priming hypothesis to different tasks is limited and that the reliability of the original findings may be more constrained than originally recognized. We make proposals for future research to examine this in future.

Required Statements during the Submission Process

1. This research is appropriate as a registered reported because the evidence speaks to an important hypothesis and theory about domain-general exploratory tendencies. Existing evidence in support of the hypothesis comes only from one research design from two separate papers by the same group of authors published in top psychology journals (Psychological Science and Journal of Experimental Psychology: General). However, there maybe an issue with publication bias, as detailed in the manuscript, where there is at least one Master's thesis that does not find support for the hypothesis but remains unpublished (in a peer reviewed journal). Thus, a registered report would test the hypothesis in such a way as to avoid issues of publication bias. Moreover, we will be using an experimental design with some minor variations to the original studies that improve and address previous shortcomings.
2. All the necessary supports (funding, facilities, ethical approvals) are already in place and the study is ready to go.
3. Upon approval of the Stage 1 manuscript, we will immediately register the approved protocol on the Open Science Framework publicly. We anticipate that following acceptance of Stage 1, the study, data analyses, and write up will be completed within 1 month and the Stage 2 manuscript will be submitted promptly thereafter (no longer than 2 months following Stage 1 approval).
4. We will share the anonymized raw data for all published results using the Open Science Framework, as we have done for the pilot study.
5. If we withdraw the paper after provisional acceptance, which we will not do, we agree that the journal can publish a short summary of the pre-registered study under a section "Withdrawn Registrations".

Final Statements

Links to the pilot data and analyses files (on the Open Science Framework) are provided in the manuscript. Our manuscript is original, not previously published, and not under concurrent consideration elsewhere. Both authors are affiliated with the Strategic Organization Design group, Department of Marketing and Management, at the University of Southern Denmark. Our contact information are as follows: Farid Anvari, faridanvari.phd@gmail.com, +45500300795; Davide Marchiori, davmar@sam.sdu.dk, +4565503694. Our mailing address is Institut for Marketing & Management, Syddansk Universitet, Campusvej 55, Odense M, 5230 Denmark.

If you have any questions, queries, or concerns, please feel free to contact the first author, Farid Anvari (faridanvari.phd@gmail.com). Thank you for considering our submission.

Sincerely,

Farid Anvari and Davide Marchiori

Appendix D

Associate Editor: 1

Comments to the Author:

The Stage 2 manuscript was evaluated by the two reviewers who assessed it at Stage 1. As you will see, both reviews are very positive while also raising a number of very thoughtful and constructive points to consider, including clarifications, additional discussion points and potential additional analyses.

I want to offer some editorial guidance here to ensure that the revision meets both the spirit and letter of the RR policy. First, please keep to a minimum any changes to the approved Stage 1 part of the manuscript. Obviously, any factual errors should be corrected, but elaboration or reframing of points of interpretation in the Introduction should be reserved for the Discussion, or at most, if vital for avoiding confusion, footnoted in the Introduction as a point raised during Stage 2 review. Second, the ideas from the reviewers concerning additional exploratory analyses strike me as interesting and well worth considering; however the inclusion of additional analyses is not required for Stage 2 RRs unless the analyses are judged necessary to justify the current conclusions. In this case, I believe that test is not met (even if the additional analyses are of interest); therefore, while I would encourage you to consider the merit of these suggestions very carefully, it is your choice as authors whether to report these additional analyses.

Provided you are able to respond thoroughly to the reviewers' comments, Stage 2 acceptance should be forthcoming without requiring further in-depth review.

****Response. We thank the editor for handling our manuscript and for the extremely helpful editorial guidance. We address each of the reviewers' points below and hope that our responses are considered thorough. When referring to the manuscript page numbers where we've added or changed any text, the reference is to the manuscript that we've submitted with track changes.*

One thing we'd like to point out is that in responding to the comments we became aware of a small error in the R code in the section where participants were being excluded based on the preregistered inclusion/exclusion criteria. This error involved a closing bracket in the wrong place, resulting in 6 participants being included in the analyses who should not have been. These 6 participants had not found the minimum required 20 words in the pretest and posttest anagram tasks. After fixing the error in the R code we reran all analyses. All conclusions remain the same. In fact, the numbers hardly changed.

The updates are made in the manuscript and can be seen on the version of the manuscript with track changes, "Participants" section on pp. 11-12 and the "Results" section on pp. 25-30. Some numbers remained the same, most changed in the second digit after the decimal (e.g., 4.29 became 4.27) and the few that changed by more than this did so by less than a full unit (e.g., -32.69 became -33.22).

Reviewer: 1

This is an impressive piece of work. The authors show in a convincing way that priming of exploration and exploitation across domains does not extend to their paradigm, which

presents some useful questions for further study in this area. Specifically, by attempting to prime exploration/exploitation from a spatial to a more internal cognitive domain (the anagram task), the authors demonstrate in a large pre-registered experiment, with some modifications from the original experiment, the absence of statistical evidence for a priming effect above a small effect size. The modifications are purposeful and invite questions about the boundary conditions for such priming.

Overall, the work is well-written, it is well-designed, the analyses are clear and straightforward, and the results are compelling. As the first author of the original work on this topic, I welcome this research and I think the authors do a good job of helping highlight the questions one would need to address to fully understand cross-domain priming. I had not fully appreciated the numerous decisions involved in evaluating this hypothesis and working through this article has helped to clarify my thinking--some of which I've written below. The authors have done an excellent job here and I think their analyses and discussion are well motivated and fair.

****Response. We thank the reviewer for the kind words. We made every effort to do the appropriate analyses to keep the conclusions as accurate as possible, given the methods and data. It's heartening that the reviewer recognized this.*

I have a few minor comments below. I say they are minor because I don't think any of them dispute the claims and interpretation made in the article. A couple at the beginning are minor corrections about prior theorizing. Otherwise, these are mostly for my own curiosity and, though probably most can be addressed with the open source data provided, I suspect some of them may help guide future research even for those who don't go delving into the data. Nonetheless, I leave it to the editor and authors to decide whether to include these in the work or not.

1. The authors say a few times that the previous work posits "a domain-general tendency to explore". I think the correct claim--the one most strongly supported by the introduction in Hills et al., 2010--is that there is a "domain general search process". That hypothesized process is likely to operate across domains and therefore be susceptible to priming. Specifically, on p. 4, line 26, and p.31 line 7, I think the wording should change to reflect that what is being primed is a domain general search process. The claim for a tendency to explore is less clear for several reasons. First, explore must be juxtaposed against the alternative: exploit, which would suggest there is not a tendency to exploit. That seems untrue in general, but it is also not the claim of the prior work.

****Response. We see the reviewer's point. To clarify what we meant, we used "domain-general tendency to explore" to refer to the idea that how people explore or exploit their environments/options, and how much they explore or exploit, will be similar across domains. Therefore, a tendency to explore also encapsulates the tendency to exploit. But of course, this wording can be misleading. In addition, we see that we were a little sloppy with our wording by stating in the introduction that the domain-general tendency to explore is a claim from the literature. Indeed, the explicit claim in the literature, as the reviewer has noted, is about a domain-general search process. What we meant to say is that a domain-general tendency to explore, as we described above, may be a*

consequent of domain-general search processes. This is because cognitive search processes calibrate the tendency to explore or to exploit (as stated in Hills et al., 2010, p. 21, “Based on Experiment 2, our interpretation of this transfer is that [...] during the foraging task a central executive search process is “tuned up” to either shows a tendency to explore or to exploit its environment”). It’s possible that we are misinterpreting Hills et al. (2010) or that we are misunderstanding how a domain-general search process would operate. We have added a footnote to the introduction to help clarify the issue for readers in order to make clear what has been stated in the literature and what we are claiming is a consequent of the literature’s claims. Specifically, the footnote on p. 3 states:

“The Stage 2 review process indicated that our wording here may be ambiguous and thus create a somewhat inaccurate representation of the claims in the literature. The past work hypothesises a domain-general cognitive *search process* that calibrates a tendency towards exploration and exploitation. A consequence of such a domain-general search process is that there is likely to be a domain-general tendency to explore, such that how much people explore and exploit, and how they explore and exploit, will be similar across domains.”

We’ve also updated the discussion section that the reviewer refers to where we now state that (the highlighted part is the new addition),

“It could be that, contrary to past theorizing (e.g., Hills & Ducas, 2012; Hills et al., 2008, 2010, 2015; Wilke et al., 2009), there is no **domain-general cognitive search process and, consequently,** no domain-general tendency to explore. But this conclusion would be too strong given the evidence. It may be that there is a **domain-general search process** but that priming doesn’t occur; the results in the published literature potentially being due to publication bias.

2. The theory behind the domain general search process is largely based on the role of executive function across domains and its putative evolution from a prior spatial search process. The notion of cross-domain priming hinges on the potential that similar processes underlying executive function are also used across domains. We do know that measures of executive function correlate with behavior across many domains, many of which (possibly all) can be characterized as search processes, or evolutionary descendants of explore/exploit processes (e.g., goal maintenance, giving up times, etc.). The question is whether the search processes can be primed across domains. In light of the present results, I see three possibilities: 1) executive function is not a single entity that is domain general, but rather a host of subprocesses that are recruited for different tasks. There is therefore nothing to prime. This is consistent with a more modular understanding of executive function--with no common elements associated with search across tasks. 2) Search processes underlying executive function are shared across domains but they aren't primable. Or 3) Search processes are shared across tasks but they rapidly adapt to new task demands and represent a small influence that is easily modified by other task demands. Indeed, on this final point, they may adapt or respond especially rapidly to situations where the 'right' level of mediating between exploration and exploitation is already known. I think the authors results are consistent with any of these three. If we incorporate the previous evidence for priming, this would favor 3). But I would not put 1) or 2) out of the question. As

the authors note, there are other reasons why we may have seen priming in the previous research.

****Response*. These are very interesting theoretical reasons that could explain our results. We've incorporated these into the relevant section of the discussion on p. 30:

"There are at least three possible explanations for our results. First, and perhaps the most parsimonious, is that exploration in one domain does not prime exploration in another domain. This explanation has several implications. It could be that, contrary to past theorizing (e.g., Hills & Ducas, 2012; Hills et al., 2008, 2010, 2015; Wilke et al., 2009), there is no domain-general cognitive search process and, consequently, no domain-general tendency to explore. That is, the cognitive search process may not be a single entity that is domain general, but rather a host of subprocesses that operate during different tasks. Therefore, there is no one thing to be primed. But this conclusion would be too strong given the evidence. It may be that there is a domain-general search process but that priming doesn't occur; the results in the published literature potentially being due to publication bias. That is, search processes may be shared across domains, but they aren't able to be primed. An alternative implication is that search processes across tasks may be shared but they rapidly adapt to new task demands so that priming doesn't occur. At this stage, all such conclusions would be somewhat premature. All we can really say is that in our study priming didn't occur at a sufficient level to be detected by the analyses. Importantly, the point estimate of the observed effect size seems very closely centered on zero ($d = -0.02$), though the confidence intervals did include effect sizes ($CI_{95\%}[-0.23, 0.19]$) that, while small by Cohen's arbitrary thresholds, may still be considered theoretically relevant."

2. A couple of times the article mentions an adaptational argument behind the basis for the priming hypothesis. This isn't clear. E.g., the work by Hills et al 2010 argues that the evolutionary basis for the shared process is likely to be one of exaptation, borrowing an existing spatial search process to search in an interior domain. I think the authors make a novel and more interesting claim but attribute it to Hills et al 2010 when they write: (p4 | 60) "Hills et al. (2010) argue that, with the structure of many search problems hinging on such persist-or-switch decisions, a domain-general mechanism mediating the explore-exploit tradeoff would be adaptive in humans." To be clear, Hills et al do argue that the existence of a search process, that can modulate between exploration and exploitation, is adaptive. But the domain generality is not argued to be adaptive, rather it is informed speculation based on the evidence for shared neuromolecular architecture in comparative biological research across domains and the evidence as explained above about executive function. I wouldn't claim these are adaptive. Indeed, I think a strongly primable search process would be maladaptive, unless we had some reason to believe that tuning the search process for one domain would inform the tuning for a subsequent very different domain. That seems unlikely. Search should be tuned to the environment in which it finds itself. Nonetheless, there may be an adaptive argument here similar to the one proposed in the article and I'm just missing it. If this can be made clearer that would be helpful.

****Response.* I think this was a misunderstanding on our part regarding the arguments made in Hills et al. (2010). Given that this is a factual error made by us (i.e., we claim that an argument was made in the literature which in fact was not) we have removed this sentence from the introduction. We thank the reviewer for this clarification.

4. I would be curious to hear more about the 50 second warning. I am curious if this 'primes' participants to signal their activity during the anagram task, which they can only do by entering words or moving forward in the task. The instructions say "if you are inactive for a substantial period of time", but I don't think the participants are told for how long, and it also says you'll be financially penalized if you are inactive for longer than the undisclosed amount of time. I'm not sure whether this would cause people to move on from one letter set to another--I suspect it would for some. Obviously, everyone is affected by this equally, but it may nonetheless be a stronger motivation to move on than any cross-domain priming effect. This isn't a criticism: for an online task (or even for a laboratory task) one needs to ensure the participants are fresh and focused. This may have been achieved in the prior studies by only having a short set of pre-test anagrams. Here it is achieved by incentivizing activity.

****Response.* The warning was simply a pop-up message that told participants that they had been inactive during the task. The reviewer is correct in so far as the participants weren't told how long they needed to be inactive to receive this warning.

5. Do people get better at the anagram task as they go on? That is, do they find more words at a faster rate or more words per letter set? Does this settle down before the post-test, suggesting they have an established expectation/strategy? Initially I thought this couldn't be the case, because the difference is negative for both conditions in Figure 2. But in fact it may be that the first four or five letter sets in the pre-test are enough to find a stable strategy (e.g., "find n words and move on"). Those using such a strategy might show a negative patch time between pre and post, assuming that the first few letter sets took longer, in order to find a reasonable point for switching. I guess I am asking why they are taking less time per letter set after the task. Are they getting better at solving anagrams, calibrating the time per letter set, or simply moving on more quickly, e.g., because they received a warning? Do people who receive a warning change their behavior by moving on more quickly?

****Response.* Although these are some potentially interesting questions, we aren't able to address some of the questions due to practical reasons. Specifically, the purpose of keeping track of the warnings was to implement the corresponding exclusion criterion. Therefore, the software keeps track of how many warnings and how long each inactivity period was, but we don't have the precise time location of the warnings with respect to the lettersets. As such, we can't do the analyses suggested by the reviewer—i.e., we can't see whether someone sped up going through lettersets after they got a warning because we don't know when they got the warning. For reference, out of the 339 participants, 83 got at least one warning. Of these 83, 66 got just 1 warning, 14 got 2 warnings, and 3 got three warnings. But of course, for all these 89, the sum of the idle times was shorter than that indicated for the exclusion.

Nevertheless, we compared the average number of words found per letterset in the pretest anagram task against the average number of words found per letterset in the posttest anagram task. This would indicate whether people were moving on more quickly past the lettersets in the posttest task. The number of words found in each letterset pretest ($M=5.4$, $SD=3.4$) was *very slightly* larger than the number of words found in each letterset posttest ($M=5.2$, $SD=3.2$). But this small mean difference ($M_{diff}=0.21$, $CI_{95\%}[-0.04,0.46]$) wasn't statistically significant ($p = .095$).

6. Because there is no travel time between patches, people will be more inclined to move to the next patch than if the travel time were longer. This is the basic prediction, over many studies, of the marginal value theorem. This isn't necessarily a problem in the present study, assuming that it takes time to warm up to a patch and this time is longer than the time it takes to find the second item after finding the first. Otherwise, the optimal strategy is always to move to the next patch immediately after finding one item. Of course, people may not realize this or act 'optimally' based on some other criterion than rate. I don't think this is a problem here, as the authors exclude people who found less than 20 words, but it might mean that people were strongly incentivized to move on from one patch to the next (in addition to the potential warning for non-activity mentioned above). The authors might want to report the average time to find the first item. More interesting would be to report the average time to find all items (first, second, third, and so on), average number of items found per patch, and how this changed across patches.

****Response.* Although these results are potentially interesting, we can't see how the average time it took participants to find each item would influence the interpretation of the confirmatory test and the conclusions that we've drawn. This is particularly the case given that all participants were given the same lettersets in the same order, meaning that both conditions would have been impacted in the same way. Hence, any priming should still show through. If there are some things related to the letterset presentation that are necessary for priming to show through, then future research can examine this, perhaps even using our data, which are shared, to explore potential hypotheses which can be tested in other confirmatory work.

7. On p. 24, l 17, the authors write that "only those who find 20 words in both pre-test and the post-test sessions will be included in the analysis." Does this mean that not all participants found 30 words in the pretest or the posttest? I may have missed it, but I guess this is because if they get through all 14 patches, then they can also complete the anagram task—or is there another reason?

****Response.* We had the requirement for participants to find a minimum of 20 words to be included due to this being a requirement used in Hills et al. (2010, Experiments 1 and 2, pp. 9 and 14, respectively; though Hills et al., 2008, didn't have this requirement). This was to make sure that we included only participants who were motivated to put in effort to find words, as opposed to rushing through the lettersets until they reached and passed the final (14th) letterset. Indeed, not all participants found 30 words. Of the 339 participants in the analyses (inc. those who found minimum of 20 words), 313 found 30 words in the pretest task and 331 found 30 words in the posttest task.

8. 150 seconds on the post-test anagram instructions seems quite long. Did many participants take longer than 20 or so seconds to read these instructions? They had already seen them once before, right? I'm guessing this is a long-tailed distribution. In a laboratory this might not be much of a problem (they might not have phones or other distractions such as email to briefly check), but I am naturally curious if this matters more for an online experiment. Nor am I sure what the right thing to do here is.

****Response.* Instruction for the post-test anagram only included a summary of the main rules of the task, with the purpose to give a parsimonious summary of the rules already seen in the pre-test anagram task. On average, participants spent 9.9 seconds (SD = 7.7 seconds) on this page of instructions. Only 21 participants spent more than 20 seconds on this page, and only 10 spent more than 30 seconds. This is therefore unlikely to matter for the interpretation of the results.

In sum, I think this is good and thorough work. Methodologically, it exemplifies good research and a state-of-the-art open science approach. Scientifically, the work raises interesting questions about the role and primability of a common search process for exploration and exploitation across domains.

****Response.* We thank the reviewer again for the kind words.

Reviewer: 2

Comments to the Author(s)

Review of "Priming exploration across domains: Does search in a spatial environment influence search in a cognitive environment?" registered report stage 2 (RSOS-201944.R3), submitted to Royal Society Open Science

Reviewed by Peter M. Todd, Indiana University Cognitive Science Program (original Reviewer 2)

This full version of the paper, reporting the pre-registered and non-pre-registered analyses on the exploration priming study results, is thorough and well-done. It is useful to see the more detailed meta-analyses with Mekern et al.'s data interpreted both in favor of and against the priming hypothesis. The modification of the spatial environments to include multiple patches is helpful for making this study design more readily comparable to that of Hills et al. 2008/2010. The additional non-pre-registered analyses, enabling consideration of the effect of search behavior (in both the meta-analysis and the current experiment) in addition to the effect of environment structure, are important for trying to understand the differing results between this study and previous research. The authors lay out a range of possible explanations and appropriate strategies for exploring them further. All of this is very beneficial for helping us gain a clearer understanding of search cognition.

****Response.* We thank the reviewer for the kind words.

One additional measure that could be informative to add to the current non-pre-registered analyses is to see whether environment structure condition differentially affects how people respond after finding or not finding a resource in the spatial search task. Searchers in

clumpy environments should stay “local” after finding a resource to increase their chances of finding other resources expected nearby, while searchers in dispersed environments should move away after finding a resource, searching more globally rather than staying local where other resources would not be expected. This was found in Hills et al. 2008 (Figure 2) in terms of turning angle, and in Wilke et al. (2015) in terms of distance jumped in a grid-search task (where win-stay versus win-shift behavior was seen). It could be that the proposed priming effect depends on resource-dependent shifting between local and global search that is common in foraging; if so, then testing for the presence of such shifting in the current data would help further constrain when priming could occur (the test would involve dividing the stepwise clustering measure used on manuscript page 26 into two measures, for those steps following finding a resource versus not finding a resource, and comparing those two measures across the two conditions).

Reference:

Wilke, A., Minich, S., Panis, M., Langen, T.A., Skufca, J.D., & Todd, P.M. (2015). A game of hide and seek: Expectations of clumpy resources influence hiding and searching patterns. *PLoS ONE*, 10(7), e0130976. doi:10.1371/journal.pone.0130976.

****Response*. This is an interesting point regarding how far away people search after they’ve found a resource and examining whether this differs between the two conditions. We therefore added the following to the manuscript’s exploratory results section (pp. 30-31):

“A reviewer suggested that a priming effect may depend on resource-dependent shifting between local and global search. That is, priming may occur only when the different environment types make people change their search strategies depending on whether they find a resource or not. Specifically, people in the clustered environments should search more locally after finding a resource, compared to after not finding a resource, to increase their chances of finding another resource tile nearby. In contrast, people in the dispersed environments should search more globally after finding a resource, as compared to after not finding a resource, to move away from the last found resource rather than staying local where resources would not be expected. This was found in Hills et al. 2008 (see their Figure 2) in terms of turning angle, and in Wilke et al. (2015) in terms of distance jumped in a grid-search task (where win-stay versus win-shift behavior was seen).

To test this, we used the sequential step clustered search measure described in the preceding paragraph to calculate the average search distance on trials after a resource was found and compared this to the average search distance on trials after a resource was *not* found. We did this separately for the dispersed and the clustered conditions. For the dispersed condition, the average distance after finding a resource tile ($M = 4.54$, $SD = 2.07$) was slightly larger than after not finding a resource ($M = 4.29$, $SD = 1.70$), $t(166) = 2.15$, $p = .033$, $d = 0.12$ $CI_{95\%}[0.01, 0.24]$. In contrast, for the clustered condition, the average distance after finding a resource was ($M = 1.49$, $SD = 0.47$) was much smaller than after not finding a resource ($M = 4.25$, $SD = 1.04$), $t(170) = 36.35$, $p < .0001$, $d = 3.36$ $CI_{95\%}[2.89, 3.83]$ —note that the degrees of freedom are correct here because there were 171 observations for this measure since one subject had a very unlucky situation in the clustered condition in which they found zero rewards in one of the environments. Therefore, we observed the

expected resource-dependent shifting between local and global search in both the clustered and dispersed environments.”

An experiment design difference that should be considered in further follow-up studies is the ability to “jump” equally easily anywhere in this grid search task, versus having to move continuously between locations in Hills et al.’s spatial search design. In the current study, local and global search “cost” roughly the same in terms of effort and time, while in Hills et al., global search takes longer to achieve, having to keep moving longer to get further across the grid (though local search takes more actions, in terms of key-presses, to turn and stay in a local area rather than moving more “straight ahead”, away from the previous location, for global search). This difference between the two spatial task versions in terms of distance and travel time may matter for possible priming.

****Response*. This is a potential explanation that falls under the second possible explanation for the divergent results that we noted in the discussion. We’ve added the following to that paragraph now, on p. 32:

“For example, in the original task used by Hills et al. people could move between patches but it was costly in terms of effort and time because it took time to move across the spatial environment. Therefore, the costs of searching locally were lower than the costs of searching globally. In contrast, in our task, the costs of local and global search were virtually equivalent since people could jump from one side of the grid to the other from one click to the next.”

One other difference that may be worth considering further, this time in the results rather than design, is that in Hills et al. 2008 the clumpy spatial environments *increased* the time spent in letter clusters while dispersed spatial environments *decreased* the time spent in letter clusters (Figure 3), but in the current study *both* environment types *decreased* the time spent in letter clusters. Why might that be?

****Response*. This is the same point raised by Reviewer 1, point 5. Please see our response there. Note, however, that it isn’t necessarily the case that the environments *caused* a decrease in time spent in the letter sets, it could be that people just got better or more used to the anagram task, having already done the task with 14 letter sets before the spatial search task. Any speculation about why is beyond the scope of our paper and wouldn’t bear on the conclusions that we draw from the confirmatory hypothesis test.

Overall, the data reported are able to test the authors’ hypothesis, which is the same as in the Stage 1 submission; the authors followed the registered experimental procedures aside from a couple of small changes that they received approval for and note in the paper; the exploratory analyses are justified, sound, and informative; and the conclusions are justified given the data.

Detailed comments by page:

Throughout: contractions (e.g. doesn’t, didn’t, we’re, they’d) should be spelled out.

****Response.* If this is an editorial/publisher requirement, we're happy to do this. However, if it's only the preference of the reviewer then we'd prefer to leave the paper as is. Contractions, at least in the first author's view, can sometimes improve readability.

Manuscript page 3 (in lower right-hand corner), lines 26-28: "there may be a domain-general tendency to explore, possibly due to common search processes being involved in different domains (e.g., Hills & Ducas, 2012; Wilke et al., 2009)." –the phrase "domain-general tendency to explore" here is ambiguous and should be clarified. It could mean that most organisms tend to explore (as opposed to exploit) in most domains they encounter (i.e., exploration is common, across species and across domains). Or it could mean that any given organism will have some individually-specific tendency to explore, from low to high, and that tendency will be similarly manifest (e.g. usually low or usually high) across the various domains that this organism explores in (i.e., individual differences in exploration tendency across domains, as assessed—but not found—by von Helversen et al. 2018 and as explored by the authors in this paper on P27L25-51). Or it could mean that search itself (including the exploration/exploitation tradeoff) is done in similar ways across species and across domains. The last interpretation seems most likely here, but given the ambiguity, this sentence should probably be reworded to be more specific, e.g. "there may be a domain-general approach to exploration and exploitation across species, possibly due to..." A similar rewording would apply to the same phrase used on P30L7-11 (in two locations).

Reference:

von Helversen, Bettina; Mata, Rui; Samanez-Larkin, Gregory R; Wilke, Andreas (2018). Foraging, exploration, or search? On the (lack of) convergent validity between three behavioral paradigms. *Evolutionary Behavioral Sciences*, 12(3):152-162.

DOI: <https://doi.org/10.1037/ebs0000121>

****Response.* This point was raised by Reviewer 1, point 1. Please refer to our response there to see how we've addressed this issue. What we refer to as the "tendency to explore" is intended to capture the latter two interpretations given by the reviewer. That is, the phrase refers to the idea that (i) how much people tend to explore (or exploit) will be similarly manifest across domains, and (ii) how people tend to explore (or exploit) will be similarly manifest across domains.

P3L28 and P30L4: "Ducas" should be "Dukas"

****Response.* These errors have been fixed.

P8L39: talking about letter-set timing here, before the cognitive task has been defined, may be confusing—can this be phrased more abstractly, e.g. in terms of patches in the cognitive task? Or describe the tasks briefly before indicating the hypotheses?

****Response.* We can see the reviewer's point. However, this section of the paper ("Overview of the Study) is specifying the statistical hypothesis, rather than the abstract theoretical hypothesis. The latter was already spelled out in the manuscript earlier. Therefore, it's important that the wording is concretely tied to the tasks used. Given that this section is merely an overview of the study, people will likely jump to the methods section to clear up any confusion.

P9L53: “averaging this across all 5 grids” – the task has not been described yet, so to make this clearer add e.g. “that each participant searched”

****Response.* We’ve added the reviewer’s suggestion to the end of that sentence (see p. 10).

P11L35-47: The participant numbers do not seem to add up properly: $400 - 47 - 2 = 351$, but 347 are indicated; and then there are 174 in the clustered condition and 171 in the dispersed condition, which is 345. And just to confirm, there were nearly twice as many female (221) as male participants (~120)?

****Response.* We thank the reviewer for pointing out these problems. These issues are a combination of ambiguous and incomplete reporting, a coding error, and a typo.

Ambiguous reporting: in the manuscript we say that “we aimed to recruit 400 participants”. But in the Prolific recruitment system participants who don’t complete the study or who are kicked out for missing the comprehension and attention checks don’t get counted towards the quota. Therefore, we ended up with 428 participant responses in total, but 34 of these were rejected for either (i) being kicked out of the study for getting the comprehension checks incorrect more than two times ($n = 13$), (ii) having abandoned the experiment ($n = 19$), or (iii) in addition, there were 2 complete responses but for which we didn’t have the Prolific ID and which we also removed ($n = 2$). Of the above 428 responses, we therefore retained $428 - 34 = 394$ participants. From these 394, we then removed participants based on the other exclusion criteria, such as for receiving the inactivity warnings and not finding the minimum required 20 words ($n = 55$; see exclusion criteria 3-7).

One error was a typo: we should have written 345 participants but had typed 347, but this point is now moot due to the coding error.

Coding error: As we described in the response to the editor, we had an error in the R code in the section where participants were being excluded based on the preregistered inclusion/exclusion criteria 3-7. This error involved a closing bracket in the wrong place, resulting in 6 participants being included in the analyses who should not have been. These 6 participants had not found the minimum required 20 words in the pretest and posttest anagram tasks. Once the coding error was fixed, applying the exclusion criteria 3-7 removed a further 55 participants from the analyses.

Thus, we ended up with a total of $(394 - 55 =)$ 339 participants ($n_{clustered} = 172$, $n_{dispersed} = 167$); age: $M = 34.6$ ($SD = 12.0$); 216 females, and 3 identifying as other. There were therefore 120 males. So, the reviewer is correct in that there were many more females in the sample.

After fixing the error in the R code we reran all analyses. All conclusions remain the same. In fact, the numbers hardly changed. Some numbers remained the same, most changed in the second digit after the decimal (e.g., 4.29 became 4.27) and the few that changed by more than this did so by less than a full unit (e.g., -32.69 became -33.22). The manuscript has been updated.

P14: Were the participants shown how many words they had generated overall, so they knew how much further they had to go (as was shown for the grid search)? Did they see the words they had already generated from the current letter-set displayed on the screen? A screen-shot of this task as well could be useful.

****Response*. During both the pre-test and post-test anagram tasks, the screen showed participants how many correct words they had formed in the task in total so far and a list of the correct words word had formed in the current letterset. We've added a footnote in the methods section for this additional detail (p. 17):

“During Stage 2 reviews we realised that some additional details of the anagram task would be beneficial. Specifically, during the pre-test and post-test anagram tasks, participants were shown the number of correct words in total they had formed thus far in the task and a list of the correct words they had formed from the current letterset.”

P23L17: “(i.e., only those who find 20 words” should be “...who find at least 20 words”

****Response*. We thank the reviewer for this point that removes the ambiguous wording. We've made the change in the manuscript.

P23L56-58: “But to complete the task participants needed to find 30 words and to be included in the analyses they needed to find 20 words.” – this is confusing—what happens if a participant does not complete the task? Do they not get paid? Do they not get to proceed to the next part of the study? Or are there no consequences? And, whatever these consequences of not completing are, their data is still analyzed if they found 20 or more words? And how is “not completing” defined—does it mean that they reach letter-cluster 14 and ask for the next cluster (which presumably does not exist) without having found 30 words?

****Response*. Participants who didn't find 30 words were still paid and could continue with the study after they'd gone through all 14 lettersets. It's simply that the task would stop for participants who found 30 words. Therefore, there were no consequences for people who didn't find the 30 words. And yes, the data were analyzed as long as the participants found a minimum of 20 words. Re the last question from the reviewer: participants were told that they'd complete the study by finding 30 words, but they weren't informed that they would also complete the study if they went through all 14 lettersets without finding 30 words. “Not completing” the study would thus be defined as actually not completing the task (i.e., they would have had to stop the study themselves before the task was completed). This isn't an important point for the analyses, given that the exclusion criteria are now clear that people would be excluded from the analyses if they didn't find at least 20 words and they would be included otherwise.

How many participants did the above classifications apply to? And how many received delay warnings or were removed because of delays or failing comprehension questions?

****Response*. Here are the number of participants who met the each of the criteria for exclusions (criteria 1-7):

1. Comprehension checks anagram task: 3
2. Comprehension checks grid search task: 10
3. Minimum of 20 words not found: 17

4 & 5. Two 50-second inactivity warnings in post-test anagram task or grid-search task: 15

6. Inactive for 100 seconds or more: 33

7. Took longer than 150 seconds on post-test anagram task instructions: 3

Importantly, some participants fit multiple of these criteria and so the total number of participants excluded due to these criteria is lower than the sum of the number of exclusions due to these criteria (see also our response to point P11L35-47, above). The number of participants excluded due purely to the exclusion criteria, without double counting participants who would have been excluded for meeting multiple criteria, is 68.

To recap, we collected 428 observations, and through criteria 1-7 we excluded 68 observations, leading to a total of 360 observations. Of these 360, 19 abandoned the experiment at some point and had to be discarded, and 2 completed the experiment, but we did not collect their Prolific IDs and for this reason were also discarded. This leads to the 339 observations we used in our analyses.

P27L9: “People in the clustered condition spent fewer minutes in each landscape (M = 5.56,” – “each landscape” should be “these landscapes”, right? That is, people presumably did not spend 5 minutes in each landscape, or else they would be spending 25 minutes on the grid search task overall (and the whole study took ~28 minutes, as indicated on P11).

****Response.* We thank the reviewer for this correction. The reviewer is correct and we’ve updated the manuscript.

In the data file, the two environment types are called “rough” and “smooth”—in the Priming Search Codebook file it would be useful to indicate the mapping of these types onto clustered and dispersed.

****Response.* We thank the reviewer again. We’ve updated the codebook file to make it clear that the smooth condition in the data is the clustered condition from the manuscript and that the rough condition in the data is the dispersed condition in the manuscript.